# VALUE MATCHING: SCALABLE AND GRADIENT-FREE REWARD-GUIDED FLOW ADAPTATION

**Cristian Perez Jensen**[*,1]  **Luca Schaufelberger**[†,1,3]  **Riccardo De Santi**[†,1,2,3]
**Kjell Jorner**[1,3]  **Andreas Krause**[1,2,3]
[1]ETH Zürich  [2]ETH AI Center  [3]NCCR Catalysis

## ABSTRACT

Adapting large-scale flow and diffusion models to downstream tasks through reward optimization is essential for their adoption in real-world applications, including scientific discovery and image generation. While recent fine-tuning methods based on reinforcement learning and stochastic optimal control achieve compelling performance, they face severe scalability challenges due to high memory demands that scale with model complexity. In contrast, methods that disentangle reward adaptation from base model complexity, such as Classifier Guidance (CG), offer flexible control over computational resource requirements. However, CG suffers from limited reward expressivity and a train-test distribution mismatch due to its offline nature. To overcome the limitations of fine-tuning methods and CG, we propose **V**alue **M**atching (VM), an online algorithm for learning the value function within an optimal control setting. VM provides tunable memory and compute demands through flexible value network complexity, supports optimization of non-differentiable rewards, and operates on-policy, which enables going beyond the data distribution to discover high-reward regions. Experimentally, we evaluate VM across image generation and molecular design tasks. We demonstrate improved stability and sample efficiency over CG and achieve comparable performance to fine-tuning approaches while requiring less than 5% of their memory usage.

## 1 INTRODUCTION

Large-scale generative foundation models have recently made remarkable progress. Among them, flow matching (Lipman et al., 2023; Albergo & Vanden-Eijnden, 2023; Liu et al., 2023a) and diffusion models (Ho et al., 2020; Sohl-Dickstein et al., 2015; Song et al., 2020) stand out for their ability to generate high-fidelity samples across a wide range of domains, including images (Ho et al.,

Table 1: Algorithm capabilities.

|  | **VM** | CG | AM | CT-PPO |
|---|---|---|---|---|
| Memory-efficient | ✓ | ✓ | ✗ | ✗ |
| Black-box rewards | ✓ | ✓ | ✗ | ✓ |
| Online data | ✓ | ✗ | ✓ | ✓ |

2020), chemistry (Hoogeboom et al., 2022), biology (Corso et al., 2023), and robotics (Chi et al., 2023). In many applications (*e.g.*, controllable image editing and drug discovery (Olivecrona et al., 2017)), it is essential to adapt these pre-trained models to downstream rewards. However, existing reinforcement learning (RL) (Zhao et al., 2025; Black et al., 2023; Fan et al., 2023; Hu et al., 2025) and stochastic optimal control (SOC) (Domingo-Enrich et al., 2025; Uehara et al., 2024; Tang, 2024; Domingo-Enrich et al., 2024) approaches update model weights via full-model backpropagation, making them increasingly memory-intensive as model sizes scale to billions of parameters (Figure 1).

Despite their promising performance (*e.g.*, Domingo-Enrich et al., 2025; Zhao et al., 2025), RL and SOC-based fine-tuning methods remain fundamentally limited by memory requirements that scale with model size. Moreover, many state-of-the-art approaches require differentiable rewards (*e.g.*, Domingo-Enrich et al., 2025), restricting applicability to black-box settings where only function evaluations are available. This is particularly limiting in applications like drug discovery, where objectives often rely on costly simulators (Bannwarth et al., 2019; Sholl & Steckel, 2009; Forli et al., 2016) or experimental measurements (Hughes et al., 2011). This raises a key question:

---

[*]Correspondence to Cristian Perez Jensen cristianpjensen@gmail.com.
[†]Equal contribution.

*How can we leverage SOC (or RL) machinery to adapt flow and diffusion models to black-box rewards with efficient resource usage and competitive performance?*

Limitations of fine-tuning methods suggest reconsidering approaches like Classifier Guidance (CG) (Dhariwal & Nichol, 2021) that offer computational advantages by avoiding updates to base model parameters and enables adaptation with black-box rewards. However, CG is an offline algorithm that trains only on samples from the pre-trained policy, leading to limited exploration beyond the training distribution. We address this shortcoming by introducing **V**alue **M**atching (VM), an online algorithm that learns the value function, enabling discovery of high-reward regions beyond the data distribution. Further, VM keeps the base generative model fixed, resulting in memory-efficient flow adaptation, and supports black-box reward functions. Lastly, we show that VM achieves comparable performance with fine-tuning methods at a significantly lower cost.

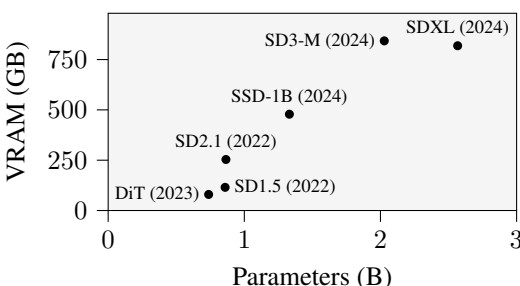

Figure 1: The recent trend toward more complex flow and diffusion generative models leads to prohibitively high fine-tuning memory (VRAM) requirements.

**Our contributions:**

- A control-theoretic viewpoint shedding light on how the offline nature of classifier guidance hinders the discovery of high-reward samples beyond the data distribution (Section 4).

- **V**alue **M**atching (VM), a theoretically-grounded online method for reward adaptation of flow models through value function learning, highlighting advantages compared to AM and CT-PPO (Section 5).

- An experimental evaluation across image and molecular generation tasks using non-differentiable rewards, showing that VM (i) achieves superior stability and sample efficiency compared to classifier guidance, (ii) reduces resource requirements by 95% relative to fine-tuning methods while maintaining comparable performance, and (iii) enables effective adaptation of molecular and large-scale text-to-image models through value networks significantly smaller than the base model (Section 6).

## 2 BACKGROUND AND NOTATION

**Flow models.** In this work, we consider the problem of adapting flow (Lipman et al., 2023; Albergo & Vanden-Eijnden, 2023; Liu et al., 2023a) and diffusion models (Ho et al., 2020; Sohl-Dickstein et al., 2015; Song et al., 2020), which have emerged as leading approaches for generative modeling across various domains, including images (Rombach et al., 2022; Peebles & Xie, 2023; Podell et al., 2024; Gupta et al., 2024; Esser et al., 2024), text (Gat et al., 2024), and molecular design (Hoogeboom et al., 2022; Dunn & Koes, 2024a;b). Typically, flow models are sampled through an ordinary differential equation (ODE) (Lipman et al., 2023), however, they can also be sampled via a stochastic differential equation (SDE) with equal time marginals (Maoutsa et al., 2020). We sample by simulating an SDE:

$$d\mathbf{x}_t = b(\mathbf{x}_t, t)\, dt + \sigma(t)\, dB_t, \quad \mathbf{x}_0 \sim p_0, \tag{1}$$

where $b : \mathbb{R}^d \times [0, 1] \to \mathbb{R}^d$ is the drift, $\sigma : [0, 1] \to \mathbb{R}^{d \times d}$ is the diffusion, and $B_t$ is a Brownian motion. For unbiased adaptation, we use a memoryless parameterization (Domingo-Enrich et al., 2025):

$$b(\mathbf{x}, t) \triangleq \frac{\dot{\alpha}_t}{\alpha_t}\mathbf{x} + \sigma^2(t)\nabla \log p_t(\mathbf{x}_t), \quad \sigma^2(t) = 2\beta_t\left(\frac{\dot{\alpha}_t}{\alpha_t}\beta_t - \dot{\beta}_t\right), \tag{2}$$

with $(\alpha_t, \beta_t)$ defined as the flow schedule (Lipman et al., 2023) and $(\dot{\alpha}_t, \dot{\beta}_t)$ being time derivatives.

**Stochastic optimal control.** For our approach, we frame KL-regularized adaptation as an SOC problem (Bellman & Dreyfus, 2015; Fleming & Rishel, 2012), a general framework that deals with optimization over stochastic processes. Specifically, we restrict ourselves to a quadratic cost control-affine Bolza problem (Bolza, 1904) on a finite time horizon $[0, 1]$ with dynamics:

$$d\mathbf{x}_t = (b(\mathbf{x}_t, t) + \sigma(t)u(\mathbf{x}_t, t))\, dt + \sigma(t)\, dB_t, \quad \mathbf{x}_0 \sim p_0. \tag{3}$$

Further, the cost functional $J$ is defined as the total cost of a control $u$ starting from a point $(\mathbf{x}, t)$, composed of a quadratic running cost $\frac{1}{2}\|u(\mathbf{x}_t, t)\|^2$ and an arbitrary terminal cost $g : \mathbb{R}^d \to \mathbb{R}$. The

*control problem* is to find the control $u$ that minimizes the cost functional $J$ at every point $(\mathbf{x}, t)$:

$$u^\star \in \arg\min_{u \in \mathcal{U}} J(u; \mathbf{x}, t) \triangleq \mathbb{E}_{p^u}\left[\frac{1}{2}\int_t^1 \|u(\mathbf{x}_s, s)\|^2 \, \mathrm{d}s + g(\mathbf{x}_1) \,\Big|\, \mathbf{x}_t = \mathbf{x}\right]. \tag{4}$$

From here, the value function is defined as the optimal value of the cost functional:

$$V(\mathbf{x}, t) \triangleq \inf_{u \in \mathcal{U}} J(u; \mathbf{x}, t) = J(u^\star; \mathbf{x}, t). \tag{5}$$

Further, $V$ can be defined through the base distribution (Domingo-Enrich et al., 2024, Appendix B):

$$V(\mathbf{x}, t) = -\log \mathbb{E}_{p_1^{\mathrm{pre}}}[\exp(-g(\mathbf{x}_1)) \mid \mathbf{x}_t = \mathbf{x}]. \tag{6}$$

## 3 PROBLEM SETTING

We consider the generative optimization problem (Li et al., 2024b; De Santi et al., 2025a;b) of adapting a pre-trained flow or diffusion model $p^{\mathrm{pre}}$ to maximize a reward function $r : \mathbb{R}^d \to \mathbb{R}$ in expectation, weighted by $\lambda \in \mathbb{R}_{\geq 0}$, and remain close to $p^{\mathrm{pre}}$ in terms of Kullback-Leiber (KL) divergence. Formally, we optimize over policies $\pi$ with induced last-timestep marginal $p_1^\pi$:

$$\arg\max_{\pi} \mathbb{E}_{p_1^\pi}[\lambda r(\mathbf{x}_1)] - D_{\mathrm{KL}}(p_1^\pi \parallel p_1^{\mathrm{pre}}) \quad \text{s.t.} \quad \mathrm{d}\mathbf{x}_t = b^\pi(\mathbf{x}_t, t) \, \mathrm{d}t + \sigma(t) \, \mathrm{d}B_t. \tag{7}$$

The optimal solution $\pi^\star$ of Problem (7) induces the tilted distribution $p_1^\star(\mathbf{x}) \propto p_1^{\mathrm{pre}}(\mathbf{x}) \exp(\lambda r(\mathbf{x}))$ if $\sigma$ follows a memoryless noise schedule (Domingo-Enrich et al., 2025). In flow models, this objective is equivalent to the following quadratic cost control-affine control problem (Tang, 2024):

$$\arg\min_{u:\mathbb{R}^d \times [0,1] \to \mathbb{R}^d} \mathbb{E}\left[\frac{1}{2}\int_t^1 \|u(\mathbf{x}_s, s)\|^2 \, \mathrm{d}s - \lambda r(\mathbf{x}_1) \,\Big|\, \mathbf{x}_t = \mathbf{x}\right] \tag{8}$$

$$\text{s.t.} \quad \mathrm{d}\mathbf{x}_t = (b^{\mathrm{pre}}(\mathbf{x}_t, t) + \sigma(t)u(\mathbf{x}_t, t)) \, \mathrm{d}t + \sigma(t) \, \mathrm{d}B_t.$$

Existing fine-tuning methods (*e.g.*, Black et al., 2023; Fan et al., 2023; Domingo-Enrich et al., 2025; Zhao et al., 2025) encounter scalability challenges because they require updating the base model parameters. To perform these updates, the gradients of the loss function must be backpropagated through the entire model. This process is highly memory-intensive, as calculating the gradients requires storing all intermediate activations of the model. As a result, the memory footprint scales with model size, posing a major bottleneck as these models grow to billions of parameters. Importantly, we focus on the black-box setting, where the reward $r$ is accessible only through function evaluations. This generalization makes the problem considerably more challenging than the setting of previous works that leverage gradient information (*e.g.*, Xu et al., 2023; Clark et al., 2024; Domingo-Enrich et al., 2025; Wang et al., 2025). Next, we introduce an approach that addresses these issues.

## 4 DISENTANGLING OPTIMIZATION THROUGH VALUE FUNCTION LEARNING

In this work, we advocate for learning the value function $V$ of Equation (5) to solve Problem (7), which offers compelling advantages over current fine-tuning methods: (1) support of non-differentiable rewards (*cf*. Xu et al., 2023; Clark et al., 2024; Domingo-Enrich et al., 2025) and (2) controllable resource usage. Once learned, the value function allows us to find the optimal control $u^\star$ through Pontryagin's minimum principle (Pontryagin, 1962), where the first-order optimality condition gives:

$$u^\star(\mathbf{x}, t) = -\sigma^\mathsf{T}(t)\nabla_{\mathbf{x}}V(\mathbf{x}, t). \tag{9}$$

A key insight for Problem (7) is that the value function retains differentiability even when the reward function is not, ensuring well-defined optimal control. Intuitively, this occurs because stochastic noise acts as a smoothing kernel; the value function $V$ at time $t$ averages over all noise realizations from $\mathbf{x}_t$ to $\mathbf{x}_1$, effectively regularizing reward discontinuities (see Figure 3). We formalize this as follows.

**Proposition 1.** *Under the memoryless noise schedule and assuming that $r$ is bounded and measurable, the value function $V$ is differentiable in $\mathbf{x}$ at $t < 1$.*

*Proof outline.* We write $V(\mathbf{x}, t) = -\log \psi(\mathbf{x}, t)$ where $\psi(\mathbf{x}, t) = \mathbb{E}_{p^{\mathrm{pre}}}[\exp(\lambda r(\mathbf{x}_1)) \mid \mathbf{x}_t = \mathbf{x}]$. Then apply the chain rule and show that $\psi > 0$ and that $\psi$ is differentiable (see Appendix A.3).

This proposition enables the optimization of non-differentiable reward functions commonly used in practical applications. It proves particularly valuable in settings where reward functions are treated as black boxes, providing only function evaluations without access to gradients or structural information. For instance, in molecular generation, reward functions often rely on external simulators (Bannwarth et al., 2019; Sholl & Steckel, 2009; Forli et al., 2016)

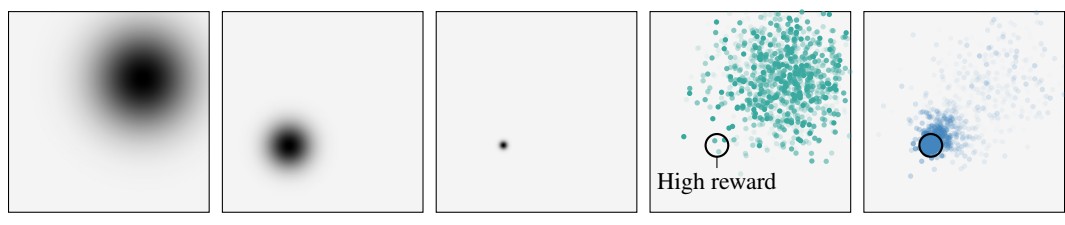

(a) Base model $p_1^{\text{pre}}$.    (b) Reward $r$.    (c) Optimal $p_1^\star$.    (d) CG training data.    (e) VM training data.

Figure 2: Training distribution evolution for VM and CG on a 2D environment, showing that VM aligns its training distribution with the optimal distribution throughout training, whereas CG does not. (2d, 2e) Points with lower opacity represent earlier training stages.

or experimental measurements (Hughes et al., 2011) that only return scalar values. In such cases, value function learning makes reward adaptation possible, as opposed to methods that rely on reward gradients (*e.g.*, Xu et al., 2023; Clark et al., 2024; Domingo-Enrich et al., 2025).

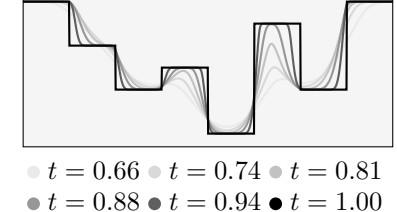

$t = 0.66$   $t = 0.74$   $t = 0.81$
$t = 0.88$   $t = 0.94$   $t = 1.00$

Figure 3: Evolution of value function for a discontinuous reward.

Beyond handling non-differentiable rewards, learning the value function disentangles reward adaptation from base model optimization. Consequently, the dominating computational cost shifts from base model training to base model inference and value function learning. This leads to *controllable resource usage* by allowing for the flexible choice of a value network architecture. As we show in Section 6, the value function can be made significantly smaller than the base model, resulting in a substantial reduction in cost, while achieving comparable performance to fine-tuning methods. Because of this significant cost reduction, the value function learning approach is highly practical for the adaptation of large-scale models where fine-tuning would be prohibitively expensive.

Having established the advantages of value function learning, we next recall how classifier guidance can be viewed as an offline algorithm to learn the value function $V$.

**Classifier guidance as offline value function learning.** As previously established by Pandey et al. (2025), Classifier guidance (CG) (Dhariwal & Nichol, 2021) admits an interpretation as value function learning. The approach involves training a classifier $p_{Y|t}(y \mid \mathbf{x})$ over noisy samples $\mathbf{x}$ at timesteps $t \in [0, 1]$ and using its log-gradient $\nabla_{\mathbf{x}} \log p_{Y|t}(y \mid \mathbf{x})$ to guide generation. This procedure can be understood as leveraging Equation (6) to solve Problem (7) with the reward function $r(\mathbf{x}) = \log p_{Y|1}(y \mid \mathbf{x})$ and value function $V(\mathbf{x}, t) = -\log p_{Y|t}(y \mid \mathbf{x})$ (see Appendix A.5 for the derivation), which in turn motivates a generalized loss function for arbitrary rewards:

$$\mathcal{L}_{\text{CG}}(\boldsymbol{\theta}; \mathbf{x}_{[0,1]}) \triangleq \frac{1}{2} \int_0^1 |\exp(-V_{\boldsymbol{\theta}}(\mathbf{x}_t, t)) - \exp(\lambda r(\mathbf{x}_1))|^2 \, dt,$$

$$d\mathbf{x}_t = b^{\text{pre}}(\mathbf{x}_t, t) \, dt + \sigma(t) \, dB_t, \quad \mathbf{x}_0 \sim p_0. \tag{10}$$

Throughout the remainder of this work, we use CG to refer to this loss formulation. Next, we discuss issues with this approach for generative optimization (Li et al., 2024b; De Santi et al., 2025a;b).

**Limitations of classifier guidance for generative optimization.** The CG formulation in Equation (10) represents an offline value function learning approach because it trains on samples from the fixed, pre-trained distribution $p_t^{\text{pre}}$ rather than the current policy distribution $p_t^u$. This offline nature is inherent to the loss formulation itself: the expectation in Equation (6) is computed over the pre-trained distribution, necessitating that the value function is trained on samples from this fixed source.

However, for generative optimization tasks such as reward maximization, we wish to sample high reward designs beyond regions of high data availability (De Santi et al., 2025a;b; 2026; Klarner et al., 2024). This objective reveals a fundamental limitation of the offline approach: a *distribution mismatch* arises between the fixed training distribution $p_t^{\text{pre}}$ and the target distribution $p_t^\star$ that the learned policy encounters during inference (see Figure 2). As the policy shifts probability mass toward higher-reward regions, the training samples from $p_t^{\text{pre}}$ become increasingly less informative, reducing sample efficiency and potentially limiting the method's ability to discover optimal solutions.

---

**Algorithm 1** Value Matching algorithm.

---

**Require:** Pre-trained base model with sampling SDE $d\mathbf{x}_t = b(\mathbf{x}_t, t)\,dt + \sigma(t)\,dB_t$, Reward function $r : \mathbb{R}^d \to \mathbb{R}$, Untrained value function approximator $V_{\boldsymbol{\theta}} : \mathbb{R}^d \times [0,1] \to \mathbb{R}$, Number of iterations $N \in \mathbb{N}$, Trajectories per iteration $m \in \mathbb{N}$, Timestep-dependent weighting $w : [0,1] \to \mathbb{R}_{>0}$.

1: **for** $N$ iterations **do**
2:  Sample $m$ trajectories under the *current policy*:

$$d\mathbf{x}_{i,t} = \left(b(\mathbf{x}_{i,t}, t) - \sigma^2(t)\nabla V_{\bar{\boldsymbol{\theta}}}(\mathbf{x}_{i,t}, t)\right)dt + \sigma(t)\,dB_t, \quad \mathbf{x}_{i,0} \sim p_0, \quad i \in [m].$$

3:  Estimate the *cost functional* for each timestep in each trajectory with $\bar{\boldsymbol{\theta}} = \texttt{stopgrad}(\boldsymbol{\theta})$:

$$\hat{J}_{i,t} = \frac{1}{2}\int_t^1 \|\sigma(s)\nabla V_{\bar{\boldsymbol{\theta}}}(\mathbf{x}_{i,s}, s)\|^2\,ds - \lambda r(\mathbf{x}_{i,1}), \quad t \in [0,1], \quad i \in [m].$$

4:  Compute the *loss function*:

$$\mathcal{L}(\boldsymbol{\theta}) = \frac{1}{2m}\sum_{i=1}^m \int_0^1 w(t) \cdot |V_{\boldsymbol{\theta}}(\mathbf{x}_{i,t}, t) - \hat{J}_{i,t}|^2\,dt.$$

5:  Make an *optimization step* with $\nabla\mathcal{L}(\boldsymbol{\theta})$.
6: **end for**

---

Additionally, the exponential terms in Equation (10) create numerical stability issues during training, due to overflows when $\lambda r(\mathbf{x}) > 90$ under 32-bit floating-point precision. This constrains the method's expressivity to small reward scalings $\lambda$, as we show in Section 6. To address these limitations, we leverage the control-theoretic viewpoint to next introduce an online value function learning approach that aligns the training and inference distributions while maintaining numerical stability.

## 5 VALUE MATCHING: SCALABLE AND GRADIENT-FREE REWARD-GUIDED ADAPTATION

We introduce **V**alue **M**atching (VM), an online method for learning the value function that overcomes a fundamental limitation of CG by training on trajectories from the current policy. As detailed in Algorithm 1, VM leverages Equations (5) and (9) to estimate the value function. This is achieved by iteratively regressing the approximator $V_{\boldsymbol{\theta}}$ onto the cost functional $J$, computed using the current policy.

$$\mathcal{L}_{\mathrm{VM}}(\boldsymbol{\theta}; \mathbf{x}_{[0,1]}) \triangleq \frac{1}{2}\int_0^1 w(t) \cdot |V_{\boldsymbol{\theta}}(\mathbf{x}_t, t) - \hat{J}(-\sigma^\mathsf{T}\nabla_{\mathbf{x}}V_{\bar{\boldsymbol{\theta}}}; \mathbf{x}_{[0,1]}, t)|^2\,dt,$$

$$\hat{J}(u; \mathbf{x}_{[0,1]}, t) \triangleq \frac{1}{2}\int_t^1 \|u(\mathbf{x}_s, s)\|^2\,ds - \lambda r(\mathbf{x}_1), \quad \bar{\boldsymbol{\theta}} = \texttt{stopgrad}(\boldsymbol{\theta}) \tag{11}$$

$$d\mathbf{x}_t = \left(b^{\mathrm{pre}}(\mathbf{x}_t, t) - \sigma^2(t)\nabla_{\mathbf{x}}V_{\bar{\boldsymbol{\theta}}}(\mathbf{x}_t, t)\right)dt + \sigma(t)\,dB_t, \quad \mathbf{x}_0 \sim p_0.$$

The VM algorithm follows a simple iterative procedure. Each iteration begins by sampling trajectories using the control policy $u(\mathbf{x}, t) = -\sigma^\mathsf{T}(t)\nabla_{\mathbf{x}}V_{\boldsymbol{\theta}}(\mathbf{x}, t)$, derived from the current value function approximator $V_{\boldsymbol{\theta}}$. These trajectories serve a dual purpose in the training loop. First, they are used to compute single-sample Monte Carlo estimates $\hat{J}_t$ of the cost functional $J$, which serves as the regression target:

$$\hat{J}_t = \frac{1}{2}\int_t^1 \sigma^2(s)\|\nabla_{\mathbf{x}}V_{\bar{\boldsymbol{\theta}}}(\mathbf{x}_s, s)\|^2\,ds - \lambda r(\mathbf{x}_1). \tag{12}$$

Here, $\bar{\boldsymbol{\theta}} = \texttt{stopgrad}(\boldsymbol{\theta})$ prevents gradients from flowing through the target, ensuring that $\hat{J}_t$ is treated as a fixed target value during backpropagation. Second, the states $\mathbf{x}_t$ along these same trajectories provide the input data for the value function. The network parameters $\boldsymbol{\theta}$ are updated by regressing the model's prediction $V_{\boldsymbol{\theta}}(\mathbf{x}_t, t)$ onto the target $\hat{J}_t$ using an $\ell_2$-loss:

$$\mathcal{L}(\boldsymbol{\theta}) = \frac{1}{2}\int_0^1 w(t) \cdot |V_{\boldsymbol{\theta}}(\mathbf{x}_t, t) - \hat{J}_t|^2\,dt. \tag{13}$$

To stabilize training under the memoryless schedule where $\sigma(t) \to \infty$ as $t \to 0$, we incorporate a weighting $w : [0, 1] \to \mathbb{R}_{>0}$. Empirically, we find the following scheme effective (see Figure 9):

$$w(t) = \frac{1}{\lambda^2 \left(1 + \frac{1}{2} \int_t^1 \sigma^2(s)\, \mathrm{d}s\right)}. \tag{14}$$

This scheme normalizes rewards by the scaling factor $\lambda$ and down-weights timesteps with high future variance. For models employing multiple schedulers, weights are averaged across schedulers.

Finally, the weights $\boldsymbol{\theta}$ are updated using a gradient descent step with $\nabla \mathcal{L}(\boldsymbol{\theta})$. For numerical implementation, the underlying SDE is simulated with the Euler-Maruyama method (Kloeden & Platen, 1992) using $T$ steps, and the integrals are approximated as Riemann sums (Anton, 1999) over the same discretization. The procedure to efficiently compute Equation (12) is outlined in Appendix B.3. In the following result, we show that this approach converges to the true value function in expectation.

**Proposition 2.** *The value function $V$ is the unique critical point of $\mathbb{E}[\mathcal{L}_{\mathrm{VM}}]$.*

*Proof outline.* We first establish that $V$ is a stationary point by computing the functional derivative of $\mathbb{E}[\mathcal{L}_{\mathrm{VM}}]$. Uniqueness then follows as a standard result in SOC (see Appendix A.4).

This proposition provides theoretical justification for using gradient-based methods to optimize $V_{\boldsymbol{\theta}}$, ensuring that VM converges to the correct value function under appropriate conditions. To build intuition for VM, we next show how it relates to current state-of-the-art fine-tuning algorithms.

**Value Matching as the gradient-free analogue of Adjoint Matching.** Conceptually, Adjoint Matching (AM) (Domingo-Enrich et al., 2025) can be understood as learning the value function gradient $\nabla_{\mathbf{x}} V$ by iteratively matching it to single-sample Monte Carlo estimates of $\nabla_{\mathbf{x}} J$. Our method represents a zeroth-order analogue, where $V$ is learned by regressing onto estimates of $J$, and the gradient is obtained via backpropagation. Thus, AM and VM are procedurally very similar.

**Value Matching simplifies continuous-time PPO.** Zhao et al. (2025) introduced a Continuous-Time Proximal Policy Optimization (CT-PPO) algorithm that learns the optimal control by iteratively alternating between training a value function and using it to optimize an actor network, starting from the pre-trained model (see Algorithm 2). We argue that by setting the actor to $s^{\mathrm{pre}}(\mathbf{x}, t) - \nabla_{\mathbf{x}} V_{\boldsymbol{\theta}}(\mathbf{x}, t)$, the actor optimization step becomes redundant and the VM algorithm emerges. This substantially simplifies the algorithm and eliminates the need for fine-tuning the base model. Moreover, VM requires fewer hyperparameters to achieve optimal performance (see Appendix E.1).

## 6 RESULTS

We now evaluate **V**alue **M**atching (VM), aiming to showcase four primary insights: (i) we verify that VM recovers the tilted distribution in an illustrative environment; (ii) we demonstrate the scalability of VM to high-dimensional image and molecular domains; (iii) in the image design task, we show VM is more sample efficient and expressive than CG; and (iv) across image and molecular design tasks, VM can reduce resource requirements by over 95% compared to fine-tuning methods, while achieving superior performance.

**VM recovers the tilted distribution with non-differentiable rewards.** To confirm that VM optimizes the intended objective, we test it in a simple one-dimensional setting, where the base distribution is a Gaussian mixture and the reward is binary and non-differentiable. We show that VM successfully converges to the tilted distribution, the optimal solution (Figure 4). Next, we consider challenging high-dimensional image and molecular generation settings.

**VM effectively adapts large-scale image generation models.** To demonstrate the general efficacy of VM on more challenging settings, we apply it to the Diffusion Transformer (DiT) (Peebles & Xie, 2023) trained on the 256×256 ImageNet dataset (Deng et al., 2009), and the text-to-image model Stable Diffusion 2 (SD2) (Rombach et al., 2022). For training prompts, we randomly selected 40k captions

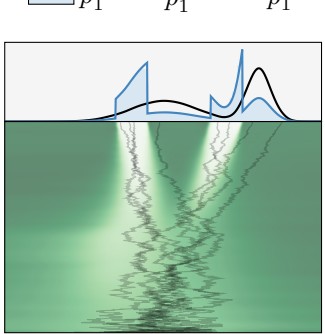

$\square\, p_1^u$ —— $p_1^{\mathrm{pre}}$ —— $p_1^{\star}$

Figure 4: 1D toy experiment.

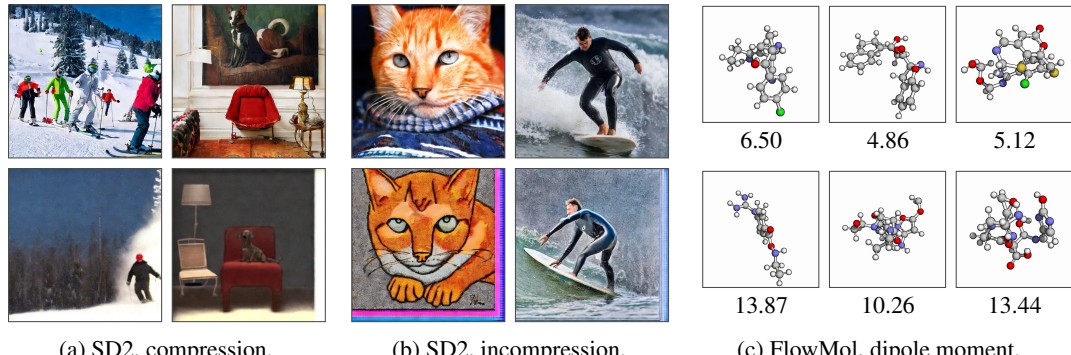

| (a) SD2, compression. | (b) SD2, incompression. | (c) FlowMol, dipole moment. |

Figure 5: Samples with same random seed. *Top*: Base model. *Bottom*: VM model.

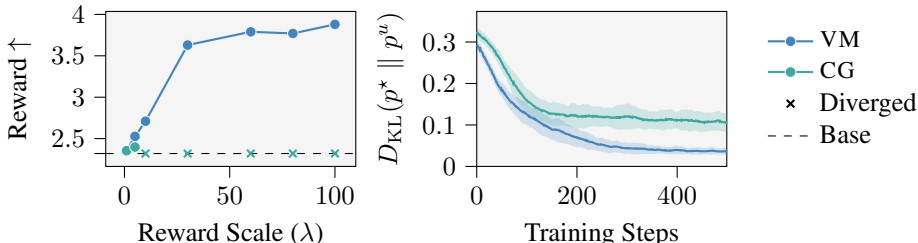

Figure 6: Comparison between VM and CG in high- and low-dimensional settings. *Left*: On the CIFAR image model with aesthetic reward, CG is unstable under moderate reward scaling. *Right*: In a simple 2D environment, VM converges significantly faster than CG to the tilted distribution in terms of KL.

from the LVIS dataset (Schuhmann & Bevan, 2023). The reward functions are compression and incompression, which correspond to minimizing and maximizing the bits per pixel (BPP) of the sample's JPEG-compressed version, at quality level 85. These rewards are non-differentiable and therefore cannot be directly optimized by gradient-based methods, such as Adjoint Matching. By learning to exploit JPEG's frequency-based method, we find that VM generates less detailed, low-frequency images under the compression reward and high-frequency Moiré patterns under the incompression reward (Appendix D). Next, we investigate the performance of VM on molecular generation.

**VM can effectively adapt molecular generation models.** In molecular design, we evaluate VM on the continuous FlowMol model (Dunn & Koes, 2024b), pre-trained on the GEOM-Drugs dataset (Axelrod & Gomez-Bombarelli, 2022). We consider two (non-differentiable) reward functions: the dipole moment computed with `GFN2-xTB` (Bannwarth et al., 2019) and QED computed with `rdkit`

Table 2: Performance of VM on FlowMol GEOM-Drugs with QED reward. Metrics computed over 10k samples.

| Method ($\lambda$) | Atom stable (%) ↑ | Stable (%) ↑ | Valid (%) ↑ | QED (0–1) ↑ |
|---|---|---|---|---|
| None (N/A) | 98.0 | 49.5 | 48.3 | 0.42 |
| VM (250) | 98.4 | 56.7 | 50.0 | 0.49 |
| VM (500) | 99.0 | 67.6 | 51.7 | 0.49 |

(Landrum et al., 2016). In both cases, molecular geometries are first relaxed using `GFN-FF` (Spicher & Grimme, 2020) before evaluating the reward. For the dipole moment reward, fragmented molecules receive a score of zero, and for the QED reward, invalid molecules are assigned a score of zero. Because atom types, edges, and formal charges are discrete and geometry relaxation introduces non-smooth transformations, both reward functions are inherently non-differentiable. Across 10K samples, VM increases the average dipole moment to 7.5 Debye from the base model's 6.4 Debye, while simultaneously reducing the fragmentation rate from 31% to 28%. We find that optimizing the dipole moment reward increases the frequency of heteroatoms and halogens, for example yielding a five-fold rise in highly electronegative fluorine (Figure 23). Further, under the QED reward, VM simultaneously improves molecular stability, validity, and QED (Table 2). Taken together, these results demonstrate that VM can successfully optimize target molecular properties without resorting to reward hacking. Next, we compare VM against classifier guidance.

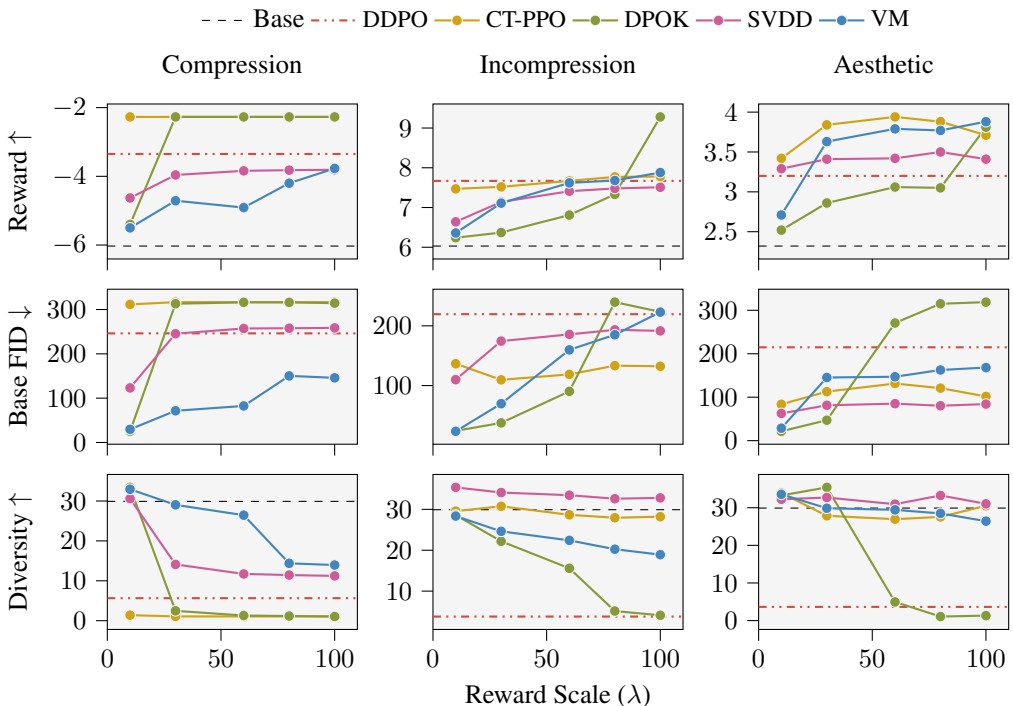

Figure 7: Comparison of VM with fine-tuning and inference-time schemes across various reward functions and scalings with CIFAR base model. Fine-tuning methods suffer from mode collapse, whereas VM does not. VM shows more predictable and stable behavior, outperforming SVDD. *Top*: Mean reward. *Middle*: FID to base model. *Bottom*: Vendi diversity with CLIP embeddings.

**VM is more sample efficient and expressive than classifier guidance.** In this comparison, we demonstrate two key advantages of VM over CG: higher reward expressivity and greater sample efficiency. To evaluate the first, we use a 32×32 CIFAR-10 base model and the LAION aesthetics reward (Schuhmann, 2022). This experiment reveals that CG becomes unstable at moderate reward scales (Figure 6, left), a significant practical limitation given that meaningful optimization often requires higher $\lambda$ values. In contrast, VM maintains stable training and leads to much higher rewards. We then evaluate sample efficiency in a one-dimensional environment by tracking the KL divergence to the optimal distribution during training. The results show VM consistently converging to superior optima (Figure 6, right), indicating a more effective use of training samples. The combination of enhanced stability and improved sample efficiency makes VM a more robust and practical alternative to CG for reward adaptation tasks, especially in resource-constrained settings. Next, we compare VM against fine-tuning methods.

**VM demonstrates superior stability compared to fine-tuning methods.** To assess how VM compares with fine-tuning approaches, we benchmark it against fine-tuning approaches CT-PPO (Zhao et al., 2025), DDPO (Black et al., 2023), and DPOK (Fan et al., 2023) on the CIFAR base model across three reward functions: compression, incompression, and aesthetics (Schuhmann, 2022), which are commonly used in prior work. Our goal is to characterize the behavior of all methods under varying reward scales $\lambda$. As shown in Figure 7, VM exhibits stronger robustness and practical reliability than competing approaches. On the compression task, all fine-tuning approaches suffer from mode collapse, whereas VM remains stable. Across the incompression and aesthetics tasks, the diversity metric indicates that both DDPO and DPOK also collapse to a single mode as $\lambda$ increases, and they consistently underperform VM in terms of reward. On the other hand, CT-PPO achieves competitive performance, but only after an extensive hyperparameter grid search (Appendix E.1), which is not necessary for VM because it has only a single hyperparameter. Moreover, VM offers more predictable and controllable behavior than CT-PPO under reward scaling. Next, we consider the computational cost of VM.

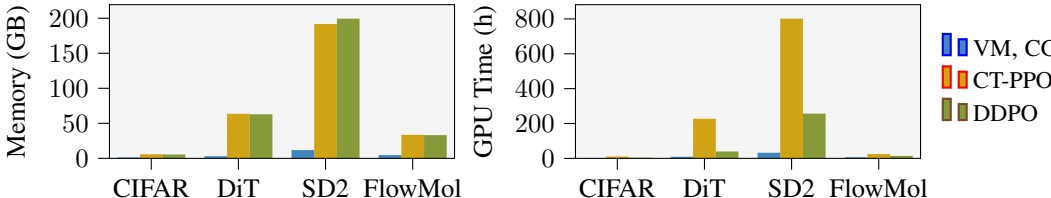

Figure 8: Resource requirements for adaptation methods across base models (16-bit, A100 GPUs). *Left*: Memory requirements in GB. *Right*: Training wall-clock time in hours.

Table 3: Scaling ablation on the CIFAR base model using the Aesthetic score as the reward function. The results are estimates over 3 runs with 1000 samples each, and the standard error is reported. Depth corresponds to the number of blocks per stage, and width is the base channel multiplier.

| Config | Depth | Width | Params (M) | Memory (GB) ↓ | Reward ↑ | Base FID ↓ | Diversity ↑ |
|--------|-------|-------|------------|---------------|----------|------------|-------------|
| None | N/A | N/A | N/A | N/A | $2.31 \pm 0.01$ | $17.23 \pm 0.17$ | $30.15 \pm 0.10$ |
| A | 1 | 32 | 0.5 | 3.2 | $3.77 \pm 0.07$ | $179.69 \pm 14.43$ | $27.44 \pm 2.72$ |
| B | 1 | 64 | 1.8 | 3.7 | $4.00 \pm 0.09$ | $208.32 \pm 4.36$ | $21.81 \pm 0.84$ |
| C | 2 | 64 | 3.8 | 4.1 | $3.67 \pm 0.09$ | $154.72 \pm 24.95$ | $28.55 \pm 3.40$ |
| D | 2 | 128 | 15.1 | 5.7 | $4.02 \pm 0.11$ | $197.60 \pm 21.35$ | $22.85 \pm 3.86$ |
| E | 3 | 128 | 23.1 | 6.6 | $2.87 \pm 0.46$ | $88.69 \pm 52.80$ | $25.61 \pm 3.43$ |
| F | 3 | 256 | 92.3 | 11.2 | $3.26 \pm 0.20$ | $94.29 \pm 21.91$ | $32.74 \pm 0.59$ |

**VM is remarkably resource-efficient.** As shown in Figure 8, VM demonstrates exceptional computational efficiency in comparison to fine-tuning methods. It requires less than 12 GB of memory across all evaluated models, whereas fine-tuning methods demand up to 250 GB for fine-tuning SD2, a reduction of over 95%. The time requirements show similar advantages: VM completes training in under 35 hours for all models, while fine-tuning methods would require up to 800 GPU-hours for SD2. This efficiency gap widens with model scale: while the resource cost for fine-tuning methods grows substantially from CIFAR to SD2, VM maintains a consistently low overhead. These results establish that VM can be orders of magnitude more efficient than fine-tuning alternatives. Next, we consider varying value network sizes.

**Small value networks are sufficient.** To understand how the complexity of the value network influences performance, we conduct a scaling ablation on the CIFAR base model using the LAION aesthetic reward (Schuhmann, 2022) with $\lambda = 100$. We evaluate six configurations (A–F), varying both the depth and width of the CNN architecture, spanning models from 0.5M to 92M parameters. We find that smaller networks are sufficient to achieve strong performance in this setting (Table 3). Increasing model size does not consistently improve reward in these experiments, so we focused on the smaller architectures in the rest of this paper. Next, we compare VM against inference-time schemes.

**Value function guidance is significantly more computationally efficient than inference-time schemes.** Unlike inference-time methods such as SVDD (Li et al., 2024a), VM adds only minimal overhead during sampling, with runtime $\mathcal{O}(T(B+G))$, where $B$ is the base model cost and $G$ the cost of computing $\nabla_{\mathbf{x}} V_{\boldsymbol{\theta}}(\mathbf{x}, t)$. This yields only a modest 1–30% increase in sampling time (Table 4). In contrast, SVDD evaluates the base model and reward for $M = 20$ candidates at each step, incurring $\mathcal{O}(TM(B+R))$ complexity and more than a 40× slowdown for compression and 600× for aesthetics.

Table 4: Inference-time overhead of VM, measured as the time to sample a batch of 128 on RTX 4090.

| Model | Base time (s) | VM time (s) |
|-------|---------------|-------------|
| CIFAR | $3.36 \pm 0.03$ | $4.47 \pm 0.01$ |
| DiT | $89.14 \pm 0.05$ | $90.37 \pm 0.08$ |
| SD2 | $122.02 \pm 0.02$ | $127.44 \pm 0.02$ |
| FlowMol | $12.77 \pm 0.14$ | $15.83 \pm 0.13$ |

Moreover, VM achieves higher aesthetic rewards while preserving diversity (Figure 7). Thus, value function guidance offers strong controllability with minimal additional inference cost, making VM far more practical than inference-time approaches.

## 7 RELATED WORK

**Reward-guided flow fine-tuning for generative optimization.** Recent work has explored fine-tuning flow and diffusion models for objectives beyond likelihood estimation, with leading approaches formulating the problem through RL and SOC frameworks. In this view, the generation process is a sequential decision problem where a policy is learned to steer the model toward desirable outcomes. An early approach, DDPO (Black et al., 2023) applies a policy gradient method to directly optimize for arbitrary rewards but often suffers from "reward collapse", where it overfits to a few high-reward samples at the cost of diversity. To counter this, DPOK (Fan et al., 2023) incorporated KL regularization to preserve diversity, though the KL term was approximated by an upper bound. A key insight was that the KL divergence can be computed with a quadratic running cost, enabling a control-theoretic interpretation. Leveraging this insight, SOCM (Domingo-Enrich et al., 2024) casts the control problem as an importance-weighted regression task. Further advancing this line, Adjoint Matching (Domingo-Enrich et al., 2025) resolved a critical value function bias in earlier methods, enabling provably unbiased reward adaptation. In an effort to address limitations of discretization, Zhao et al. (2025) introduced a continuous-time RL framework. Complementary to these methods, LaSRO (Jia et al., 2025) improves reward adaptation in few-step diffusion models by learning a differentiable latent-space surrogate reward. Recent work also renders it possible to maximize rewards while preserving information from $p^{\mathrm{pre}}$ more generally than KL, as well as enabling risk-averse and risk-sensitive reward optimization (De Santi et al., 2025a; Wang et al., 2026). Our work advances this research line by introducing an algorithm that preserves the online nature of control-theoretic schemes, while lowering the memory requirements significantly, thereby easing its practical adoption.

**Classifier(-free) guidance.** A widely used alternative to fine-tuning is to steer the generative process at inference time. Classifier guidance (Dhariwal & Nichol, 2021) leverages the gradients of a separately trained classifier to push the sampling trajectory toward samples that exhibit desired attributes. To eliminate the need for an external model, classifier-free guidance (Ho & Salimans, 2022) modifies the training of the generative model itself to learn both a conditional and an unconditional distribution. At inference, the model is guided by amplifying the difference between the two, effectively steering generations toward the desired condition. These guidance mechanisms are foundational in diffusion model research, and improving upon them has become an active field of study (*e.g.*, Karras et al., 2024; Sadat et al., 2024; 2025; Rajabi et al., 2025; Perez Jensen & Sadat, 2025). In this work, we establish a connection by showing that VM can be viewed as an online generalization of classifier guidance.

**Inference-time schemes beyond guidance.** Another family of methods performs reward optimization at inference-time through local, step-wise decisions. Many of these approaches can be understood as approximating an optimal denoising process by leveraging the pre-trained model as a look-ahead function to predict future rewards (Uehara et al., 2025). For instance, at each denoising step, methods like SVDD (Li et al., 2024a) and SCG (Huang et al., 2024) evaluate multiple candidate states and select the next state based on these predictions, employing strategies such as resampling or greedy selection. In contrast to such local methods, FlowGrad (Liu et al., 2023b), D-Flow (Ben-Hamu et al., 2024), and OC-Flow (Wang et al., 2025) adopt a global perspective, optimizing the entire trajectory by leveraging reward gradients. The downside of these methods is a substantially increased wall-clock time for generation. VM does not suffer from this by amortizing the optimization cost during training.

## 8 CONCLUSION AND OUTLOOK

We introduce **V**alue **M**atching (VM), a scalable and efficient online method for adapting pre-trained flow models to arbitrary reward functions. Drawing from fundamental insights in optimal control theory, VM learns the value function, which yields several key advantages. First, the resource requirements are controllable by a flexible choice of the value network architecture. Second, by learning the value function, VM naturally handles non-differentiable rewards, a crucial capability for black-box optimization.

Our experiments on image and molecular generation tasks demonstrate these benefits empirically. VM reduces memory and compute requirements by up to 95% compared to fine-tuning methods while achieving comparable performance. Furthermore, VM shows higher reward expressivity and proves more sample efficient than classifier guidance. By providing a theoretically grounded and practical framework for reward-guided adaptation, VM opens up promising opportunities for future research, such as applying VM to more complex, real-world problems.

## REPRODUCIBILITY STATEMENT

We provide comprehensive details to ensure the reproducibility of our work. For all algorithms introduced, pseudocode is included, and benchmarks are performed against existing, publicly documented methods. In Appendix B.1, we detail the value network architectures used in this work. Further, in Appendix B.3, we show how to compute the integrals in practice. Moreover, in Appendix C, we give descriptions of the evaluation metrics, including how they are computed. Lastly, in Appendix E.1, we report the hyperparameters used for the CT-PPO experiments. For other baselines (DDPO, DPOK, and SVDD), we use the hyperparameters specified in their respective paper.

## ACKNOWLEDGMENTS

This publication was made possible by the ETH AI Center doctoral fellowship to Riccardo De Santi. The project has received funding from the Swiss National Science Foundation under NCCR Catalysis (grant number 180544 and 225147) and NCCR Automation (grant agreement 51NF40 180545). This work was supported by an ETH Zurich Research Grant.

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

# APPENDICES

## CONTENTS

# A  PROOFS

## A.1  ASSUMPTIONS

**Assumption 1.** $a(t) \triangleq \sigma(t)\sigma^{\mathsf{T}}(t)$ is uniformly elliptic.

**Assumption 2.** The base drift $b : \mathbb{R}^d \times [0,1] \to \mathbb{R}^d$ is Lipschitz continuous in $\mathbf{x}$ and continuous in $t$.

**Assumption 3.** The norm of the base drift $\|b\|$ is bounded.

**Assumption 4.** The reward function $r : \mathbb{R}^d \to \mathbb{R}$ is bounded.

## A.2  USEFUL LEMMAS

**Lemma 1** (Application of the Feynman-Kac formula (Kac, 1949)). *Let $V$ be the value function defined in Equation (5), then:*

$$V(\mathbf{x}, t) = -\log \mathbb{E}_{p^{\text{pre}}}[\exp(\lambda r(\mathbf{x}_1)) \mid \mathbf{x}_t = \mathbf{x}]. \tag{15}$$

**Lemma 2** (Friedman (1975); Chapter 6, Theorem 4.5). *Under Assumptions 1 to 3, the transition density $p_{s|t}(\mathbf{y} \mid \mathbf{x})$ of the uncontrolled SDE satisfies the following upper bound on its norm for $0 \le t < s \le 1$:*

$$\|\nabla_{\mathbf{x}} p_{s|t}(\mathbf{y} \mid \mathbf{x})\| \le C(s-t)^{-\frac{d+1}{2}} \exp\left(-c\frac{\|\mathbf{y} - \mathbf{x}\|^2}{s-t}\right), \tag{16}$$

*where $C, c > 0$ are constants and $\mathbf{x}, \mathbf{y} \in \mathbb{R}^d$. Further, $\nabla_{\mathbf{x}} p_{s|t}(\mathbf{y} \mid \mathbf{x})$, $\nabla_{\mathbf{x}}^2 p_{s|t}(\mathbf{y} \mid \mathbf{x})$, and $\partial_t p_{s|t}(\mathbf{y} \mid \mathbf{x})$ are uniformly continuous.*

**Lemma 3** (Fleming & Soner (2006), Chapter 5, Theorem 9.1). *Consider the following Hamilton-Jacobi-Bellman equation:*

$$-\partial_t W(\mathbf{x}, t) + \mathcal{H}(\mathbf{x}, t, \nabla_{\mathbf{x}} W(\mathbf{x}, t), \nabla_{\mathbf{x}}^2 W(\mathbf{x}, t)) = 0, \tag{17}$$

*where in our case the Hamiltonian, $\mathcal{H}$, is:*

$$\mathcal{H}(\mathbf{x}, t, \mathbf{p}, \mathbf{A}) = -\frac{1}{2}\operatorname{tr}(a(t)\mathbf{A}) - \langle b(\mathbf{x}, t), \mathbf{p} \rangle + \frac{1}{2}\|\sigma^{\mathsf{T}}(t)\mathbf{p}\|^2. \tag{18}$$

*Assume Assumptions 1 to 4. Let $W$ be a bounded viscosity subsolution and $V$ be a bounded viscosity supersolution. Then,*

$$\sup_{(\mathbf{x}, t) \in \mathbb{R}^d \times [0,1]} (W(\mathbf{x}, t) - V(\mathbf{x}, t)) = \sup_{\mathbf{x} \in \mathbb{R}^d} (W(\mathbf{x}, 1) - V(\mathbf{x}, 1)). \tag{19}$$

**Lemma 4** (Uniqueness). *Under the assumptions of Lemma 3, the viscosity solution to Equation (17) is unique.*

*Proof.* We show this result by a comparison principle. Assume that Equation (17) has two viscosity solutions $V_1$ and $V_2$ with terminal condition $V_1(\mathbf{x}, 1) = V_2(\mathbf{x}, 1) = -\lambda r(\mathbf{x})$. Since they are viscosity solutions, they are also viscosity sub- and supersolutions. By their terminal condition, we know that:

$$\sup_{\mathbf{x} \in \mathbb{R}^d} (W(\mathbf{x}, 1) - V(\mathbf{x}, 1)) = \sup_{\mathbf{x} \in \mathbb{R}^d} -\lambda r(\mathbf{x}) + \lambda r(\mathbf{x}) = 0. \tag{20}$$

We will first show that $V_1 \le V_2$. Apply Lemma 3 with $W = V_1$ and $V = V_2$, then we have:

$$V_1(\mathbf{x}, t) - V_2(\mathbf{x}, t) \le \sup_{(\mathbf{x}, t)} (V_1(\mathbf{x}, t) - V_2(\mathbf{x}, t)) = \sup_{\mathbf{x} \in \mathbb{R}^d} (V_1(\mathbf{x}, 1) - V_2(\mathbf{x}, 1)) = 0. \tag{21}$$

As such we have $V_1(\mathbf{x}, t) \le V_2(\mathbf{x}, t)$ for all $(\mathbf{x}, t)$.

Now we show that $V_2 \le V_1$. Again, apply Lemma 3 with $W = V_2$ and $V = V_1$, then we have:

$$V_2(\mathbf{x}, t) - V_1(\mathbf{x}, t) \le \sup_{(\mathbf{x}, t)} (V_2(\mathbf{x}, t) - V_1(\mathbf{x}, t)) = \sup_{\mathbf{x} \in \mathbb{R}^d} (V_2(\mathbf{x}, 1) - V_1(\mathbf{x}, 1)) = 0. \tag{22}$$

Thus we also have $V_2(\mathbf{x}, t) \le V_1(\mathbf{x}, t)$ for all $(\mathbf{x}, t)$.

In conclusion, we have $V_1 - V_2 = 0$, meaning that they are equal. $\square$

Putting it all together, we have that the following HJB equation has a unique solution:

$$\partial_t W(\mathbf{x}, t) + \frac{1}{2}\operatorname{tr}(a(t)\nabla_{\mathbf{x}}^2 W(\mathbf{x}, t)) + \langle b(\mathbf{x}, t), \nabla_{\mathbf{x}} W \rangle - \frac{1}{2}\|\sigma^{\mathsf{T}}(t)\nabla_{\mathbf{x}} W\|^2 = 0,$$
$$W(\mathbf{x}, 1) = -\lambda r(\mathbf{x}). \tag{23}$$

### A.3 PROPOSITION 1

*Under Assumptions 1 to 4, the value function $V : \mathbb{R}^d \times [0,1] \to \mathbb{R}$ defined in Equation (5) is differentiable in $\mathbf{x}$ for $t < 1$*

*Proof.* Let $t < 1$. Define $\psi(\mathbf{x}, t) \triangleq \mathbb{E}_{p^{\mathrm{pre}}}[\exp(\lambda r(\mathbf{x}_1)) \mid \mathbf{x}_t = \mathbf{x}]$. Then from Lemma 1, we have $V(\mathbf{x}, t) = -\log \psi(\mathbf{x}, t)$. Thus, it suffices to show that (1) $\psi > 0$ and (2) $\psi$ is differentiable in $\mathbf{x}$:

1. We assume that $r$ is bounded, so $\exp(\lambda r(\mathbf{x})) > 0$. Hence, $\psi(\mathbf{x}, t) > 0$.

2. Writing $\psi$ as an integral we have:

$$\psi(\mathbf{x}, t) = \int \exp(\lambda r(\mathbf{y})) p_{1|t}(\mathbf{y} \mid \mathbf{x}) \, \mathrm{d}\mathbf{y}. \tag{24}$$

Using that $r$ is bounded such that $\exp(\lambda r) < M$ for some $M$ and Lemma 2, we can show that the gradient norm of the integrand is dominated by an integrable function:

$$\|\nabla_{\mathbf{x}} \exp(\lambda r(\mathbf{y})) p_{1|t}(\mathbf{y} \mid \mathbf{x})\| = \exp(\lambda r(\mathbf{y})) \|\nabla_{\mathbf{x}} p_{1|t}(\mathbf{y} \mid \mathbf{x})\| \tag{25}$$

$$\leq MC(1-t)^{-\frac{d+1}{2}} \exp\left(-c\frac{\|\mathbf{y} - \mathbf{x}\|^2}{1-t}\right), \tag{26}$$

where $M, C, c, d > 0$ are constants. Thus, we can differentiate under the integral:

$$\nabla \psi(\mathbf{x}, t) = \int \exp(\lambda r(\mathbf{y})) \nabla_{\mathbf{x}} p_{1|t}(\mathbf{y} \mid \mathbf{x}) \, \mathrm{d}\mathbf{y}. \tag{27}$$

Further using Lemma 2, the transition density is continuously differentiable in $\mathbf{x}$. Thus, $\psi$ is differentiable.

This fails for $t = 1$ since $r$ might be non-differentiable and we have $V(\mathbf{x}, 1) = -\lambda r(\mathbf{x})$. In conclusion, by the chain rule:

$$\nabla V(\mathbf{x}, t) = -\frac{\nabla \psi(\mathbf{x}, t)}{\psi(\mathbf{x}, t)} \tag{28}$$

Therefore, $V$ is continuously differentiable in $\mathbf{x}$ for $t < 1$. $\qquad\square$

### A.4 PROPOSITION 2

*The value function $V$ is the unique critical point of $\mathbb{E}[\mathcal{L}_{\mathrm{VM}}]$.*

*Proof.* Let $W : \mathbb{R}^d \times [0,1] \to \mathbb{R}$ be a value function approximator and denote $\bar{W} = \mathtt{stopgrad}(W)$ where the argument of $\mathtt{stopgrad}$ is treated as constant w.r.t. differentiation. In this proof, assume that any trajectory $\mathbf{x}_{[0,1]}$ is sampled from the current policy without gradients w.r.t. weights:

$$\mathrm{d}\mathbf{x}_t = \left(b(\mathbf{x}_t, t) - \sigma(t)\sigma^{\intercal}(t)\nabla\bar{W}(\mathbf{x}_t, t)\right)\mathrm{d}t + \sigma(t)\,\mathrm{d}B_t. \tag{29}$$

*Critical point.* In order to find the critical points, we will derive the functional derivative of $\mathbb{E}[\mathcal{L}_{\mathrm{VM}}]$. Let $C : \mathbb{R}^d \times [0,1] \to \mathbb{R}$ be an arbitrary function, then:

$$\frac{\mathrm{d}}{\mathrm{d}\epsilon}\mathbb{E}[\mathcal{L}_{\mathrm{VM}}(W + \epsilon C; \mathbf{x}_{[0,1]})]\bigg|_{\epsilon=0} \tag{30}$$

$$= \frac{\mathrm{d}}{\mathrm{d}\epsilon}\mathbb{E}\left[\frac{1}{2}\int_0^1 w(t) \cdot \left|(W + \epsilon C)(\mathbf{x}_t, t) - \hat{J}\left(-\sigma^{\intercal}\nabla\bar{W}; \mathbf{x}_{[0,1]}, t\right)\right|^2 \mathrm{d}t\right]\bigg|_{\epsilon=0} \tag{31}$$

$$= \mathbb{E}\left[\frac{1}{2}\int_0^1 w(t) \cdot \frac{\mathrm{d}}{\mathrm{d}\epsilon}\left|(W + \epsilon C)(\mathbf{x}_t, t) - \hat{J}\left(t - \sigma^{\intercal}\nabla\bar{W}; \mathbf{x}_{[0,1]}, t\right)\right|^2\bigg|_{\epsilon=0} \mathrm{d}t\right] \tag{32}$$

$$= \mathbb{E}\left[\int_0^1 C(\mathbf{x}_t, t) \cdot w(t) \cdot \left(W(\mathbf{x}_t, t) - \hat{J}\left(-\sigma^{\intercal}\nabla\bar{W}; \mathbf{x}_{[0,1]}, t\right)\right)\mathrm{d}t\right] \tag{33}$$

Using the tower property of expectation:

$$= \mathbb{E}\left[\int_0^1 C(\mathbf{x}_t, t) \cdot w(t) \cdot \left(W(\mathbf{x}_t, t) - \mathbb{E}\left[\hat{J}\left(-\sigma^\mathsf{T}\nabla\bar{W}; \mathbf{x}_{[0,1]}, t\right) \,\middle|\, \mathbf{x}_t\right]\right) \mathrm{d}t\right]. \tag{34}$$

So, the functional derivative is:

$$\frac{\delta}{\delta W}\mathbb{E}[\mathcal{L}_{\mathrm{VM}}(W)(\mathbf{x}, t)] = w(t) \cdot \left(W(\mathbf{x}, t) - \mathbb{E}\left[\hat{J}\left(-\sigma^\mathsf{T}\nabla\bar{W}; \mathbf{x}_{[0,1]}, t\right) \,\middle|\, \mathbf{x}_t = \mathbf{x}\right]\right). \tag{35}$$

Thus, any critical point (a point where the functional derivative equals zero) must satisfy:

$$W^\star(\mathbf{x}, t) = \mathbb{E}\left[\hat{J}\left(-\sigma^\mathsf{T}\nabla W^\star; \mathbf{x}_{[0,1]}, t\right) \,\middle|\, \mathbf{x}_t = \mathbf{x}\right] \tag{36}$$

$$= \mathbb{E}\left[\frac{1}{2}\int_t^1 \|\sigma^\mathsf{T}(s)\nabla W^\star(\mathbf{x}_s, s)\|^2 \,\mathrm{d}s - \lambda r(\mathbf{x}_1) \,\middle|\, \mathbf{x}_t = \mathbf{x}\right]. \tag{37}$$

By plugging Equation (9) into Equation (5), we know that the value function can be written as:

$$V(\mathbf{x}, t) = J(u^\star; \mathbf{x}, t) \tag{38}$$

$$= J(-\sigma^\mathsf{T}\nabla V; \mathbf{x}, t) \tag{39}$$

$$= \mathbb{E}\left[\frac{1}{2}\int_t^1 \|\sigma^\mathsf{T}(s)\nabla V(\mathbf{x}_s, s)\|^2 \,\mathrm{d}s - \lambda r(\mathbf{x}_1) \,\middle|\, \mathbf{x}_t = \mathbf{x}\right]. \tag{40}$$

Therefore $V$ is a critical point of $\mathbb{E}[\mathcal{L}_{\mathrm{VM}}]$.

*Uniqueness.* As shown, a critical point $W$ must satisfy the fixed-point:

$$W(\mathbf{x}, t) = \mathbb{E}\left[\frac{1}{2}\int_t^1 \|\sigma^\mathsf{T}(s)\nabla W(\mathbf{x}_s, s)\|^2 \,\mathrm{d}s - \lambda r(\mathbf{x}_1) \,\middle|\, \mathbf{x}_t = \mathbf{x}\right], \tag{41}$$

where the expectation is over trajectories from the controlled SDE:

$$\mathrm{d}\mathbf{x}_s = (b(\mathbf{x}_s, s) - a(s)\nabla W(\mathbf{x}_s, s))\,\mathrm{d}s + \sigma(s)\,\mathrm{d}B_s, \quad \mathbf{x}_t = \mathbf{x}. \tag{42}$$

By the Feynman-Kac formula, $W$ satisfies the following PDE:

$$\partial_t W + \langle b - a\nabla W, \nabla W\rangle + \frac{1}{2}\mathrm{tr}\left(a\nabla^2 W\right) + \frac{1}{2}\|\sigma^\mathsf{T}\nabla W\|^2 = 0, \quad W(\mathbf{x}, 1) = -\lambda r(\mathbf{x}). \tag{43}$$

Noticing that $\langle\nabla W, a\nabla W\rangle = \|\sigma^\mathsf{T}\nabla W\|^2$, the PDE simplifies to:

$$\partial_t W + \langle b, \nabla W\rangle + \frac{1}{2}\mathrm{tr}\left(a\nabla^2 W\right) - \frac{1}{2}\|\sigma^\mathsf{T}\nabla W\|^2 = 0, \quad W(\mathbf{x}, 1) = -\lambda r(\mathbf{x}). \tag{44}$$

Using Lemma 4, we know that this HJB equation has a unique solution. This concludes the proof of Proposition 2: $V$ is the unique critical point of $\mathbb{E}[\mathcal{L}_{\mathrm{VM}}]$. $\qquad\square$

## A.5 DERIVATION OF CLASSIFIER GUIDANCE VALUE FUNCTION

In this setting, we have $r(\mathbf{x}) = \log p_{Y|1}^{\mathrm{pre}}(y \mid \mathbf{x})$ for some class label $y$ and $\lambda = 1$. From Lemma 1, we have:

$$V(\mathbf{x}, t) = -\log\mathbb{E}_{p^{\mathrm{pre}}}[\exp(\lambda r(\mathbf{x}_1)) \mid \mathbf{x}_t = \mathbf{x}] \tag{45}$$

$$= -\log\mathbb{E}_{p^{\mathrm{pre}}}[p^{\mathrm{pre}}(y \mid \mathbf{x}_1) \mid \mathbf{x}_t = \mathbf{x}] \tag{46}$$

$$= -\log\int p_{1|t}^{\mathrm{pre}}(\mathbf{x}_1 \mid \mathbf{x})p_{Y|1}^{\mathrm{pre}}(y \mid \mathbf{x}_1) \,\mathrm{d}\mathbf{x}_1 \tag{47}$$

We have $y \perp \mathbf{x}_t \mid \mathbf{x}_1$, so by the chain rule:

$$= -\log\int p_{1,Y|t}^{\mathrm{pre}}(\mathbf{x}_1, y \mid \mathbf{x}) \,\mathrm{d}\mathbf{x}_1 \tag{48}$$

By marginalization:

$$= -\log p_{Y|t}^{\mathrm{pre}}(y \mid \mathbf{x}). \tag{49}$$

This concludes the derivation of the statement.

# B    EXPERIMENTAL DETAILS

All VM experiments employ the Adam optimizer (Kingma & Ba, 2014) with a learning rate of $1 \times 10^{-4}$, a batch size of 128, and 100 SDE discretization steps. Unless otherwise specified, image experiments utilize a 1.8M-parameter CNN and molecular experiments employ a 2.5M-parameter graph neural network (GNN) to parameterize the value function approximator $V_{\theta}$. Additionally, to normalize the CG loss function (as we do for VM by adding the $1/\lambda^2$ term to $w(t)$), we normalize $\mathcal{L}_{\text{CG}}$ by dividing it by $\exp(2\lambda)$.

## B.1    VALUE NETWORK ARCHITECTURES

**Convolutional neural network.**    We employ a standard CNN architecture consisting of an input convolution, three downsampling stages, an adaptive average pool, and a final linear head. Timesteps are embedded using a sinusoidal timestep embedder (Vaswani et al., 2017). Each downsampling stage comprises two layers with the following structure: convolution with a 3×3 kernel $\rightarrow$ group normalization (Wu & He, 2018) $\rightarrow$ FiLM (Perez et al., 2018) to incorporate timestep information $\rightarrow$ sigmoid linear unit (Hendrycks & Gimpel, 2016) activation function. Finally, the input is added residually and the result is downsampled using blur pool (Zhang, 2019). The convolutional layers in the downsampling stages use a base hidden dimensionality of 64, which doubles at each stage.

**Graph neural network.**    The GNN architecture follows a similar design to the CNN architecture (excluding downsampling), where we replace the input convolution with a linear layer, convolutions with graph convolutions (Kipf & Welling, 2017), and group normalization with layer normalization (Ba et al., 2016). We utilize all node information available from the FlowMol model: atom position, atom type, and formal charge. We also incorporate edge data by linearly transforming the edge features and adding the mean of all incoming edge features to the node features after the input linear layer. Each block uses a hidden dimensionality of 256 across 6 stages.

## B.2    EFFICIENTLY COMPUTING REWARDS

To efficiently compute reward functions for latent diffusion models, we decode samples individually. This approach significantly reduces VRAM requirements, as decoded samples are typically very large. We find that this strategy does not result in substantially increased wall-clock time.

## B.3    EFFICIENTLY COMPUTING THE COST FUNCTIONAL ESTIMATE

We discretize the time horizon into $T$ evenly spaced points. On this discretization, we perform the Euler-Maruyama method for sampling. Thus, at each step, the gradient $\nabla_{\mathbf{x}}V$ is computed. Based on this gradient, we compute the running cost at every step:

$$L_t = \frac{1}{2}\|\sigma^{\mathsf{T}}(t)\nabla_{\mathbf{x}}V(\mathbf{x}_t, t)\|^2. \tag{50}$$

At the final time step, the reward $R = r(\mathbf{x}_1)$ is received. The estimated cost functional $\hat{J}_t$ is then computed by summing the running costs from time $t$ onward and subtracting the scaled terminal reward:

$$\hat{J}_t = \frac{1}{T}\sum_{\tau=t}^{T} L_{\tau/T} - \lambda R. \tag{51}$$

In total, computing the cost functional estimate involves computing $T$ $d$-dimensional norms and adding $T$ scalars using a reverse cumulative sum.

## B.4 WEIGHTING FUNCTIONS

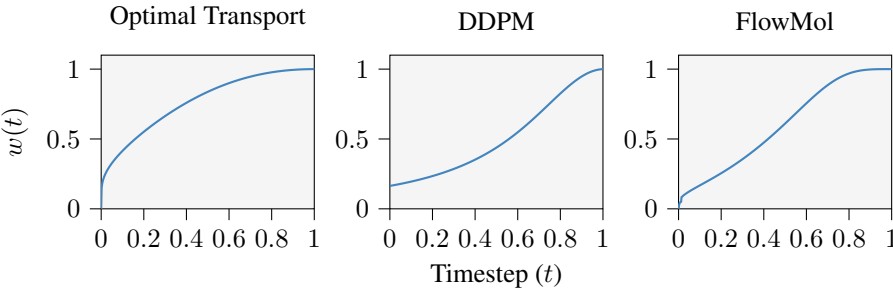

Figure 9: Weighting function $w(t)$ under various $(\alpha_t, \beta_t)$-schedules, $\lambda = 1$, and the memoryless noise schedule.

Figure 9 displays the weighting function defined in Equation (14) for the schedules of models considered in this work. As can be seen, it down-weights earlier timesteps, which intuitively have the highest variance.

## C  EVALUATION METRICS

Throughout this work, we employ three key metrics to assess performance: average reward, Fréchet inception distance (FID) (Heusel et al., 2017), and Vendi diversity (Friedman & Dieng, 2023; Pasarkar & Dieng, 2024). The average reward measures an algorithm's ability to exploit the reward function effectively, while FID relative to the base model captures the extent of deviation from the base model required to achieve this performance. Vendi diversity quantifies the variety within generated samples. Our objective is to achieve high reward and diversity while maintaining low FID. However, an inherent trade-off exists between reward optimization and sample diversity. In this section, we detail the computation of these metrics. For each metric, we assume access to a dataset of $n$ samples.

### C.1  FRÉCHET INCEPTION DISTANCE

FID (Heusel et al., 2017) is computed by first embedding each data point through a pre-trained Inception network (Szegedy et al., 2015) and extracting feature activations from the final layer. The Fréchet distance computes the means $(\boldsymbol{\mu}_1, \boldsymbol{\mu}_2)$ and covariance matrices $(\boldsymbol{\Sigma}_1, \boldsymbol{\Sigma}_2)$ of both datasets, then calculates:

$$d_F(\mathbf{X}_1, \mathbf{X}_2) \triangleq \|\boldsymbol{\mu}_1 - \boldsymbol{\mu}_2\|^2 + \mathrm{tr}\Big(\boldsymbol{\Sigma}_1 + \boldsymbol{\Sigma}_2 - 2(\boldsymbol{\Sigma}_1\boldsymbol{\Sigma}_2)^{1/2}\Big). \tag{52}$$

Typically, $\mathbf{X}_1$ represents a reference dataset and $\mathbf{X}_2$ contains samples from the generative model. In this work, however, we set the reference dataset to samples from the base model and $\mathbf{X}_2$ to samples from the reward-adapted model. This provides a measure of how much the reward-adapted version has deviated from the base model.

### C.2  VENDI SCORE

The Vendi score (Friedman & Dieng, 2023; Pasarkar & Dieng, 2024) is a diversity metric that requires only a positive semi-definite similarity function $k : \mathcal{X} \times \mathcal{X} \to \mathbb{R}$ with $k(\mathbf{x}, \mathbf{x}) = 1$ for all $\mathbf{x} \in \mathcal{X}$. It computes pairwise similarities between all samples and organizes them into a matrix $\mathbf{K} \in \mathbb{R}^{n \times n}$ where $k_{ij} = k(\mathbf{x}_i, \mathbf{x}_j)$. The Vendi score is defined as the exponential of the entropy of the eigenvalues of $\mathbf{K}/n$:

$$\mathrm{VS}_k(\{\mathbf{x}_1, \ldots, \mathbf{x}_n\}) \triangleq \exp\left(-\sum_{i=1}^{n} \lambda_i \log \lambda_i\right). \tag{53}$$

In this work, we employ the following similarity function:

$$k(\mathbf{x}, \mathbf{y}) = \langle \mathrm{clip}(\mathbf{x}), \mathrm{clip}(\mathbf{y}) \rangle, \tag{54}$$

where $\mathrm{clip}(\cdot)$ represents a CLIP image encoder (Radford et al., 2021) that produces normalized embeddings.

# D    SAMPLES AND TRAINING CURVES

The plotted costs reflect the deviation of the fine-tuned model from the base model; they correspond to the KL divergence between the base and controlled processes $p^u$ (Domingo-Enrich et al., 2025):

$$D_{\mathrm{KL}}\big(p^u(\mathbf{x}_{[0,1]}) \,\big\|\, p^{\mathrm{pre}}(\mathbf{x}_{[0,1]})\big) = \mathbb{E}_{p^u}\left[\frac{1}{2}\int_0^1 \|u(\mathbf{x}_t, t)\|^2\,\mathrm{d}t\right] \tag{55}$$

## D.1    DIFFUSION TRANSFORMER

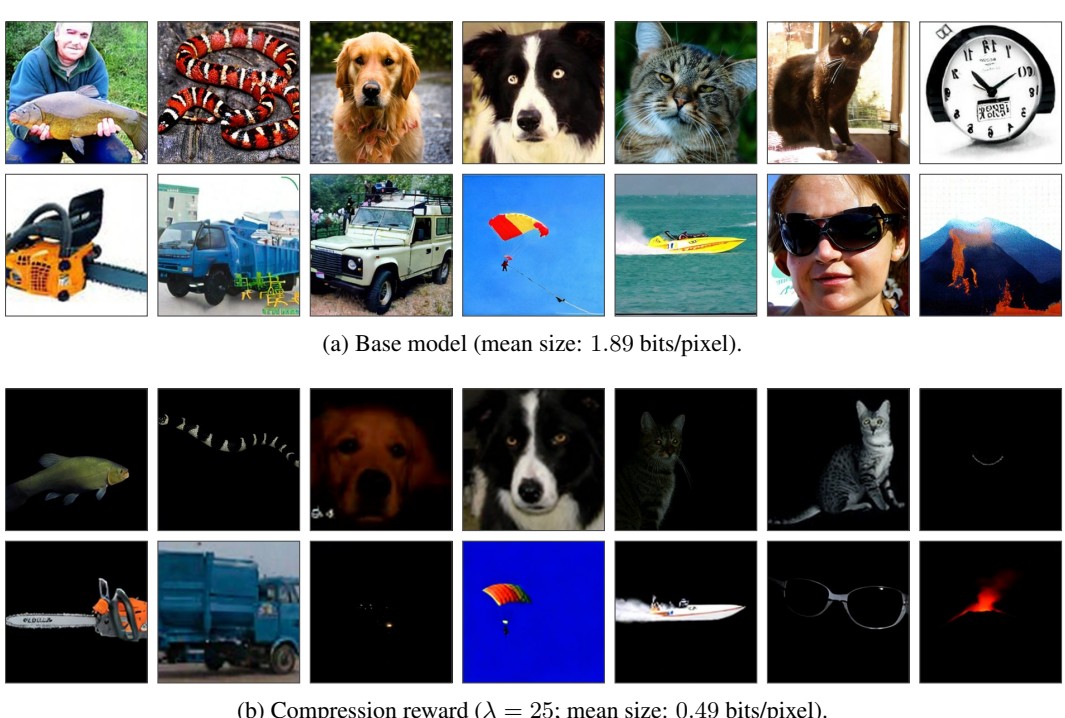

(a) Base model (mean size: 1.89 bits/pixel).

(b) Compression reward ($\lambda = 25$; mean size: 0.49 bits/pixel).

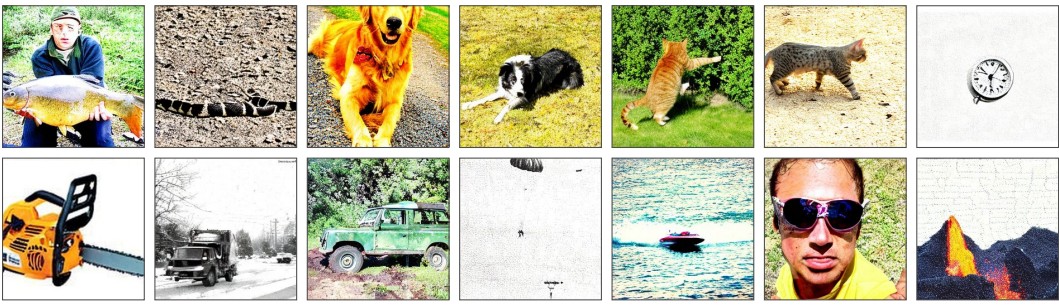

(c) Incompression reward ($\lambda = 25$; mean size: 3.31 bits/pixel).

Figure 10: Samples from Diffusion Transformer generated under the same random seed. Inference with CFG weight 2, whereas training was done without CFG.

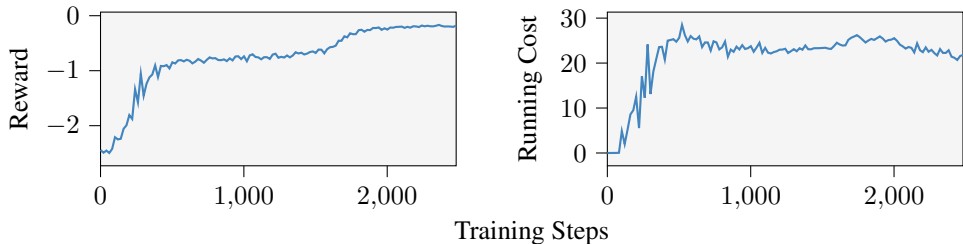

Figure 11: Training curves for VM on the DiT model with compression reward ($\lambda = 25$).

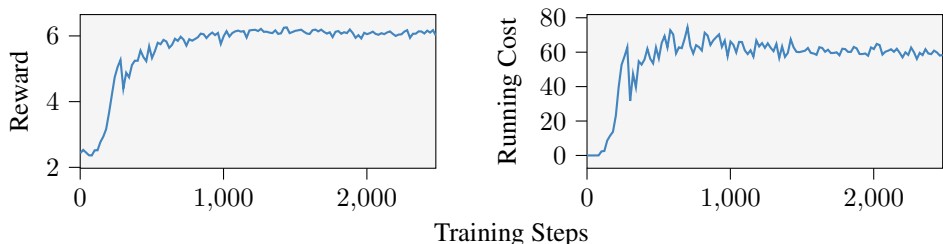

Figure 12: Training curves for VM on the DiT model with incompression reward ($\lambda = 25$).

## D.2 STABLE DIFFUSION 2

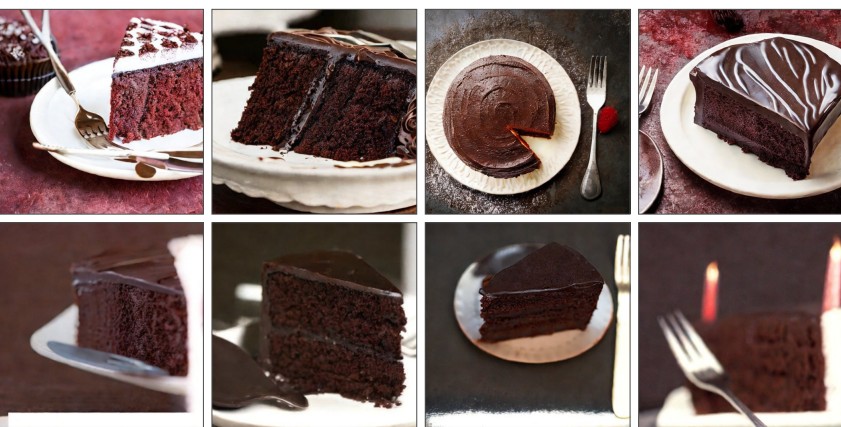

Figure 13: Prompt: *A chocolate cake on a plate with decorative pattern, a fork beside it, giving off a sense of indulgence or celebration.* Reward: Compression ($\lambda = 2500$). CFG weight: 4.0.

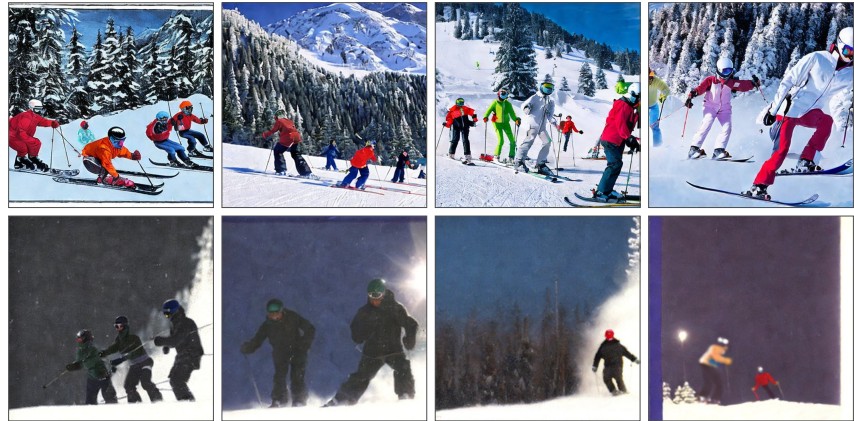

Figure 14: Prompt: *Skiing scene with multiple individuals dressed in ski gear, engaging in skiing activities amidst snowy surroundings, suggesting a resort or slope ambiance.* Reward: Compression ($\lambda = 2500$). CFG weight: 4.0.

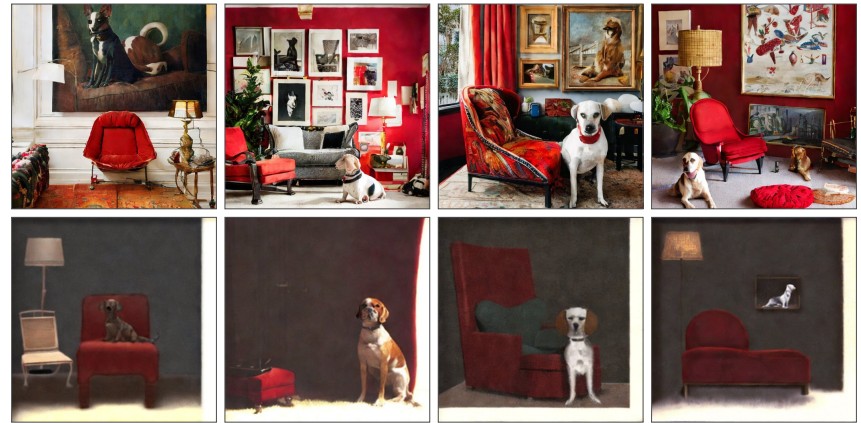

Figure 15: Prompt: *Dog seated on a red lounge chair in a cozy, sophisticated room with a painting, various decorations, and multiple lampshades while wearing a collar.* Reward: Compression ($\lambda = 2500$). CFG weight: 4.0.

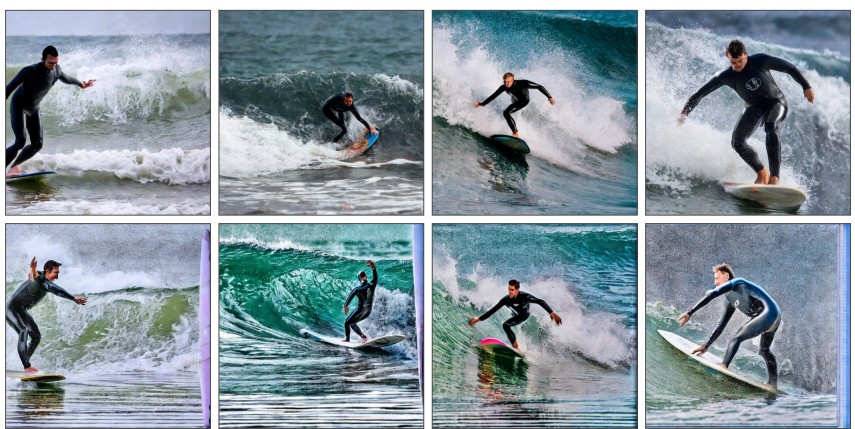

Figure 16: Prompt: *A surfer in a wet suit performs a carving turn by a pier, on a beach break with no other surfers or boats present.* Reward: Incompression ($\lambda = 2500$). CFG weight: 4.0.

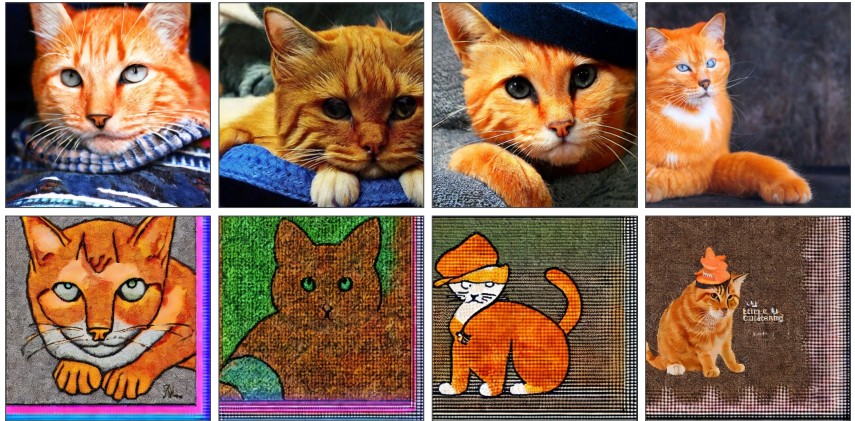

Figure 17: Prompt: *An orange cat with a blue hat featuring a logo, resting on a dark-colored background.* Reward: Incompression ($\lambda = 2500$). CFG weight: 4.0.

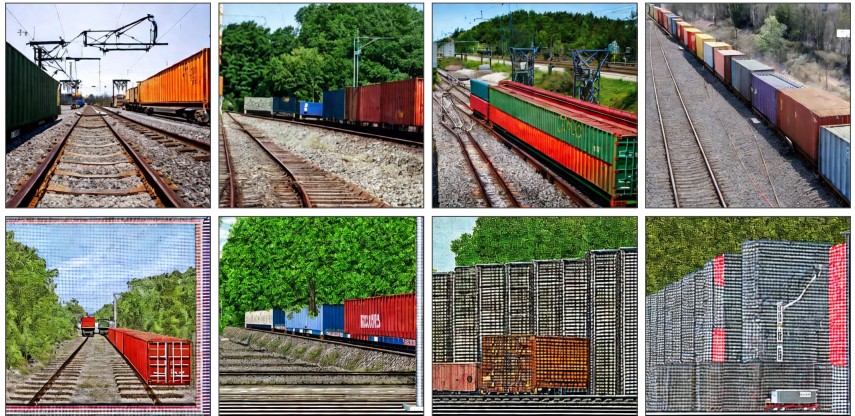

Figure 18: Prompt: *A freight train with cargo containers passes through a railroad crossing.* Reward: Incompression ($\lambda = 2500$). CFG weight: 4.0.

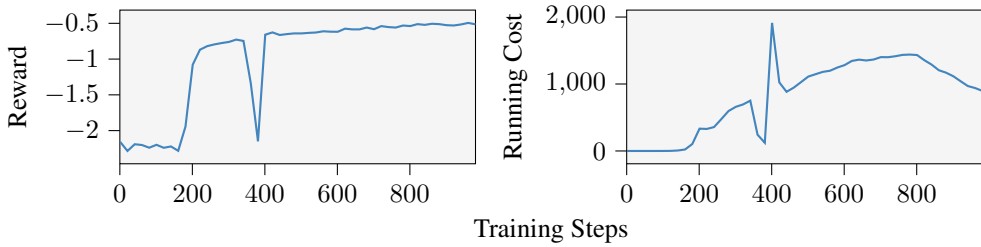

Figure 19: Training curves for VM on the SD2 model with compression reward ($\lambda = 2500$).

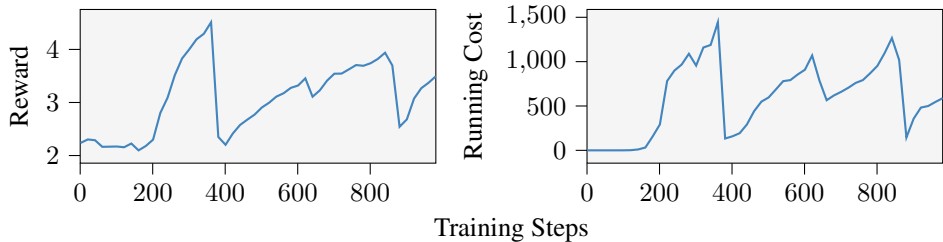

Figure 20: Training curves for VM on the SD2 model with incompression reward ($\lambda = 2500$).

## D.3 GEOM-DRUGS

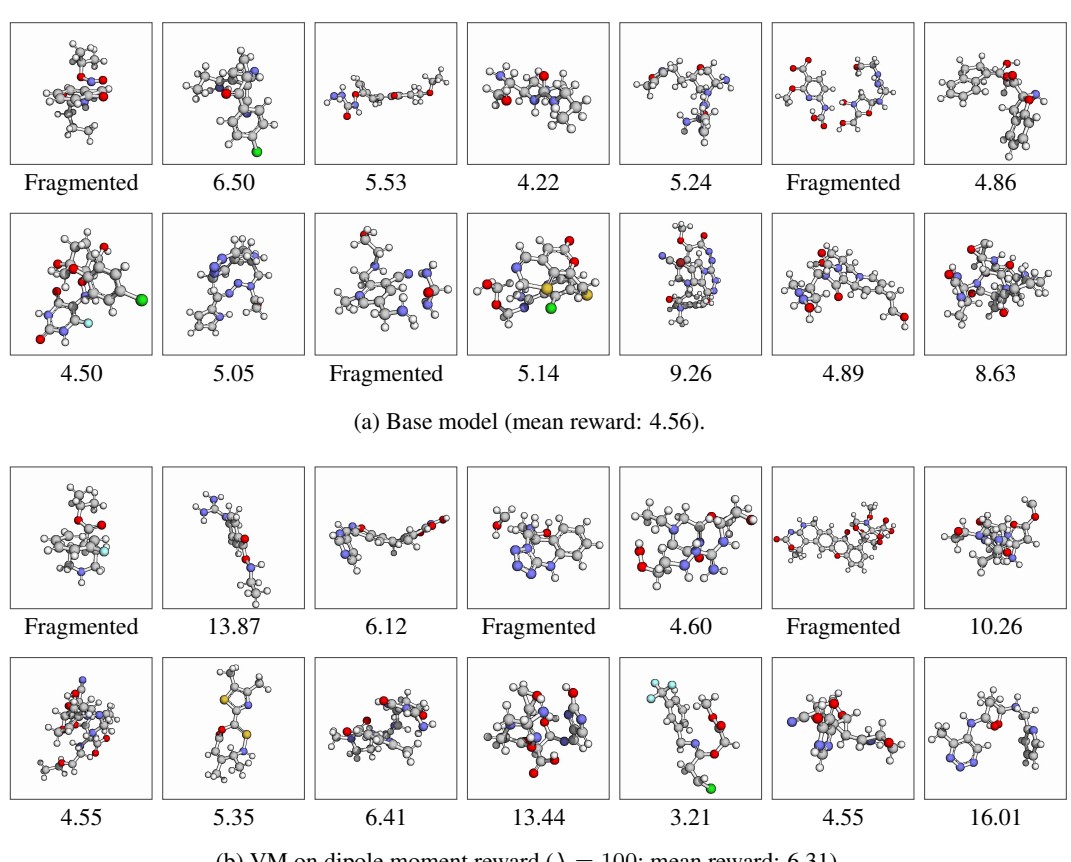

Figure 21: Samples from the continuous GEOM-Drugs FlowMol base and VM model under the same random seed.

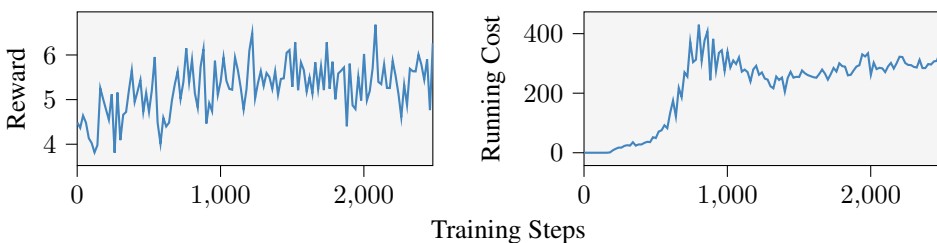

Figure 22: Training curves for VM on the continuous GEOM-Drugs FlowMol model with dipole moment reward ($\lambda = 100$).

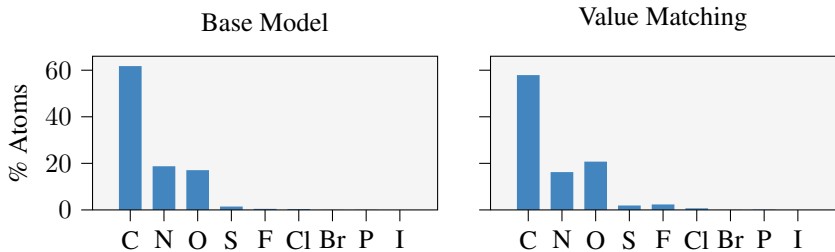

Figure 23: Atom type frequency distributions for molecules generated by FlowMol base model and VM with dipole moment reward ($\lambda = 100$). Results shown for 10K generated samples.

# E   BASELINES

## E.1   CONTINUOUS-TIME PROXIMAL POLICY OPTIMIZATION (CT-PPO)

---

**Algorithm 2** Continuous-Time PPO (CT-PPO) algorithm.

---

**Require:** Initial policy parameters $\boldsymbol{\theta}$ (pre-trained), Initial value function parameters $\phi$, Reward function $r : \mathbb{R}^d \to \mathbb{R}$, Number of iterations $N \in \mathbb{N}$, Training steps per iteration $K \in \mathbb{N}$, Trajectories per iteration $m \in \mathbb{N}$, Batch size $B \in \mathbb{N}$, Exploration level $\sigma \in \mathbb{R}_{>0}$, Scaling parameter $\eta \in \mathbb{R}_{>0}$, Clipping parameter $\epsilon \in \mathbb{R}_{>0}$.

1: **for** $N$ iterations **do**
2:    Fix current policy $\bar{\boldsymbol{\theta}} = \boldsymbol{\theta}$.
3:    Sample $m$ trajectories under the current policy:

$$\mathrm{d}\mathbf{x}_t = \left( \frac{\dot{\alpha}_t}{\alpha_t} \mathbf{x}_t + \sigma^2(t)\mathbf{a}_t \right) \mathrm{d}t + \sigma(t)\,\mathrm{d}B_t, \quad \mathbf{a}_t = s_{\bar{\boldsymbol{\theta}}}(\mathbf{x}_t, t).$$

4:    Compute returns:

$$R_t = r(\mathbf{x}_1) - \frac{1}{2\lambda} \int_t^1 \sigma^2(s) \| s_{\bar{\boldsymbol{\theta}}}(\mathbf{x}_s, s) - s^{\mathrm{pre}}(\mathbf{x}_s, s) \|^2 \, \mathrm{d}s.$$

5:    Initialize dataset $\mathcal{D}_V = \{(t, \mathbf{x}_t, R_t)\}_{t \in [0,1]}$ with all trajectories.
6:    **repeat** $K$ **times**
7:        Sample $\mathcal{B} \subset \mathcal{D}_V$ with batch-size $B$.
8:        Compute loss:

$$\mathcal{L}(\phi) = \frac{1}{B} \sum_{(t, \mathbf{x}_t, R_t) \in \mathcal{B}} (V_\phi(\mathbf{x}_t, t) - R_t)^2.$$

9:        Make an optimization step with $\nabla \mathcal{L}(\phi)$.
10:    **end repeat**
11:    Sample exploration noise $\boldsymbol{\epsilon}_t \sim \mathcal{N}(\mathbf{0}, \mathbf{I})$ independently for each timestep and trajectory.
12:    Compute pseudo-samples and advantages:

$$\tilde{\mathbf{a}}_t = \mathbf{a}_t + \sigma \boldsymbol{\epsilon}_t$$
$$q_t = \frac{1}{\eta} \big( V_\phi(\mathbf{x}_t + \eta \sigma^2(t) \boldsymbol{\epsilon}_t, t) - V_\phi(\mathbf{x}_t, t) \big).$$

13:    Initialize dataset $\mathcal{D}_\pi = \{(t, \mathbf{x}_t, \tilde{\mathbf{a}}_t, q_t)\}_{t \in [0,1]}$ with all trajectories.
14:    **repeat** $K$ **times**
15:        Sample $\mathcal{B} \subset \mathcal{D}_\pi$ with batch-size $B$.
16:        Compute likelihood ratio:

$$\rho_t^{\boldsymbol{\theta}} = \frac{\pi_{\boldsymbol{\theta}}(\tilde{\mathbf{a}}_t \mid \mathbf{x}_t, t)}{\pi_{\bar{\boldsymbol{\theta}}}(\tilde{\mathbf{a}}_t \mid \mathbf{x}_t, t)}, \quad \pi_{\boldsymbol{\theta}}(\mathbf{a} \mid \mathbf{x}, t) = \mathcal{N}(\mathbf{a}; s_{\boldsymbol{\theta}}(\mathbf{x}, t), \sigma \mathbf{I}).$$

17:        Compute loss:

$$\mathcal{L}(\boldsymbol{\theta}) = \frac{1}{B} \sum_{(t, \mathbf{x}_t, \tilde{\mathbf{a}}_t, q_t) \in \mathcal{B}} \min \big\{ \rho_t^{\boldsymbol{\theta}} q_t, \mathrm{clip}(\rho_t^{\boldsymbol{\theta}}, 1 - \epsilon, 1 + \epsilon) q_t \big\}.$$

18:        Make an optimization step with $\nabla \mathcal{L}(\boldsymbol{\theta})$.
19:    **end repeat**
20: **end for**

---

We set $K = \lceil m/B \rceil$ such that each point is seen once. For the actor and critic, we use learning rates $3 \times 10^{-5}$ and $1 \times 10^{-6}$, respectively. For data collection, we standardize the number of trajectories and batch size using $m = 512$, $B = 128$, and $N = 250$. This configuration processes 128K trajectories during training, consistent with other methods in this work.

**Hyperparameter ablations.** To ensure a fair comparison, we conduct comprehensive hyperparameter optimization for CT-PPO through an extensive grid search over the clipping parameter $\epsilon$, exploration level $\sigma$, and scale $\eta$. Specifically, we conducted a grid search on $(\epsilon, \sigma, \eta) \in \{0.05, 0.1, 0.2\} \times \{0.01, 0.1, 0.2\} \times \{0.001, 0.005, 0.01\}$. While this additional tuning effort could be considered part of CT-PPO's computational overhead, it ensures optimal performance for our evaluation. In contrast, both VM and AM do not have any hyperparameter search cost.

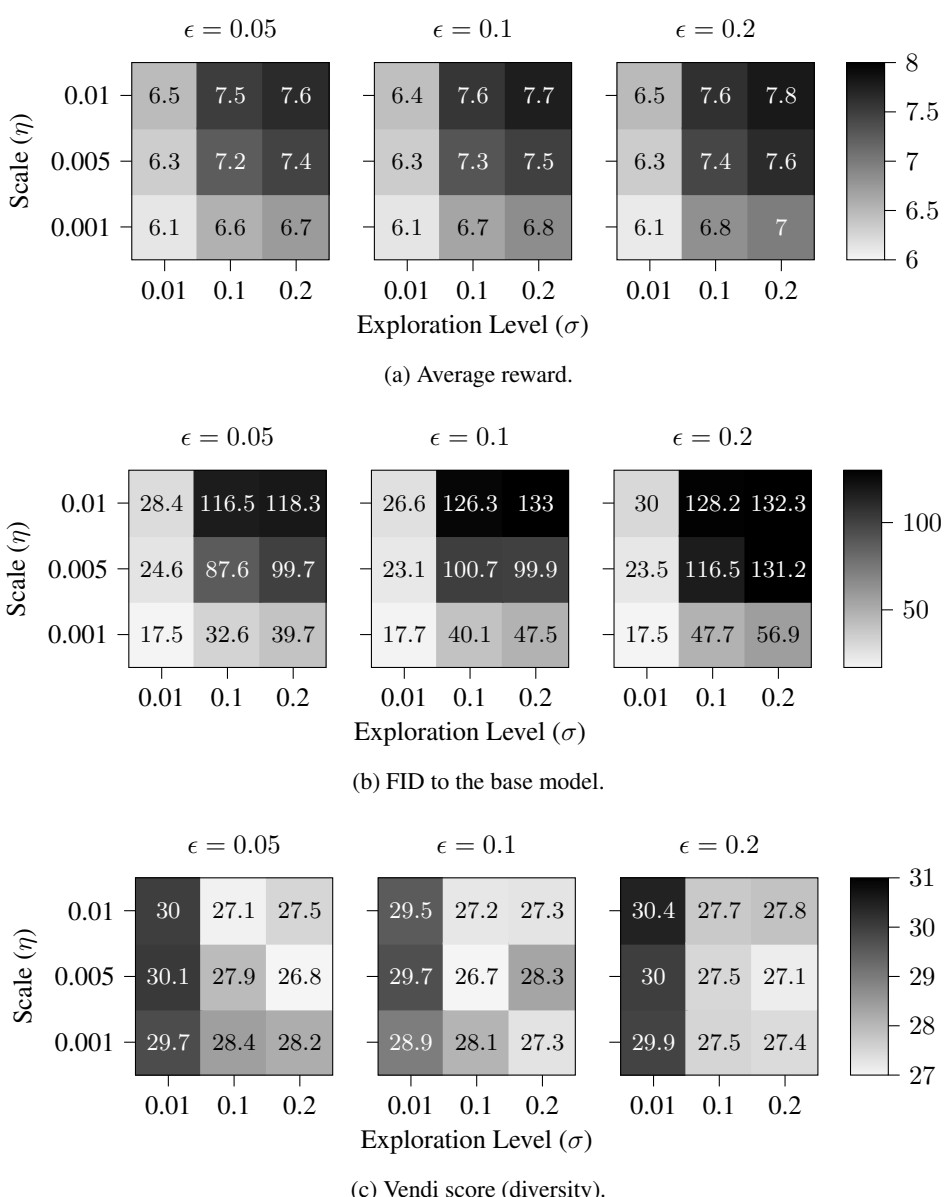

(a) Average reward.

(b) FID to the base model.

(c) Vendi score (diversity).

Figure 24: Base model: CIFAR. Reward: Incompression ($\lambda = 100$).

# F  FLOW MODELS

For completeness, in this section we provide an overview of flow matching (Lipman et al., 2023; Albergo & Vanden-Eijnden, 2023; Liu et al., 2023a) and how diffusion models (Ho et al., 2020; Sohl-Dickstein et al., 2015; Song et al., 2020) can be viewed as an instance of it. Further, we show how to sample flow matching models through an SDE with equivalent time marginals. Lastly, we show how to sample flow matching or diffusion models from a common perspective through the score function.

## F.1  FLOW MATCHING

Given a source distribution $p$ and a target distribution $q$, the flow matching framework aims to solve the flow matching problem:

*Find the velocity field $v : \mathbb{R}^d \times [0, 1] \to \mathbb{R}^d$ generating marginal distributions $p_t$, where $p_0 = p$ and $p_1 = q$.*

The flow matching framework solves this problem by the following steps:

1. Identify a known source distribution $p$ and unknown target distribution $q$, of which we have finite samples.
2. Define a probability path $p_t$ that interpolates $p_0 = p$ and $p_1 = q$.
3. Learn the velocity field by a neural network $v_{\boldsymbol{\theta}}$.
4. Sample the learned model by solving an ODE:

$$\mathrm{d}\mathbf{x}_t = v_{\boldsymbol{\theta}}(\mathbf{x}_t, t)\,\mathrm{d}t. \tag{56}$$

In general, we could use a coupled data distribution $\mathbf{x}_0, \mathbf{x}_1 \sim p_{0,1}$, however, we will only be considering the case where $\mathbf{x}_0 \sim \mathcal{N}(\mathbf{0}, \mathbf{I}_d)$. Further, $q$ is unknown, but we do assume that we have access to a dataset of samples from this distribution. *E.g.*, we might want to model a distribution of images and take the 32 ×32 CIFAR-10 dataset (Krizhevsky et al., 2009) as samples from this distribution.

Next, we need to define a probability path $\{p_t\}_{t \in [0,1]}$ that interpolates between $p_0 = p$ and $p_1 = q$. This is done by a conditional strategy, which involves defining $p_{t|1}$. We can then construct the marginal probability path by:

$$p_t(\mathbf{x}) = \int p_{t|1}(\mathbf{x} \mid \mathbf{x}_1) q(\mathbf{x}_1)\,\mathrm{d}\mathbf{x}_1. \tag{57}$$

We will consider an affine parameterization of the conditional probability path:

$$p_{t|1}(\mathbf{x} \mid \mathbf{x}_1) = \mathcal{N}\big(\mathbf{x}; \alpha_t \mathbf{x}_1, \beta_t^2 \mathbf{I}_d\big), \tag{58}$$

where $\alpha_t, \beta_t : [0, 1] \to [0, 1]$ are smooth functions satisfying $\alpha_0 = \beta_1 = 0$, $\alpha_1 = \beta_0 = 1$, and $\dot{\alpha}_t > 0 > \dot{\beta}_t$ for $t \in (0, 1)$. (The dot-notation denotes the time-derivative.) We can sample from this distribution as follows:

$$\mathbf{x}_{t|1} = \alpha_t \mathbf{x}_1 + \beta_t \mathbf{x}_0, \quad \mathbf{x}_0 \sim p_0. \tag{59}$$

Commonly, the optimal transport schedule is used where $\alpha_t = t$ and $\beta_t = 1 - t$.

Differentiating w.r.t. $t$ gives the associated marginal velocity field:

$$v(\mathbf{x}, t) = \mathbb{E}[\dot{\alpha}_t \mathbf{x}_1 + \dot{\beta}_t \mathbf{x}_0 \mid \mathbf{x}_t = \mathbf{x}]. \tag{60}$$

Thus, we can train using the flow matching loss:

$$\mathcal{L}_{\mathrm{FM}}(\boldsymbol{\theta}) \triangleq \mathbb{E}_{t,\mathbf{x}_t}\big[\|v_{\boldsymbol{\theta}}(\mathbf{x}_t, t) - v(\mathbf{x}_t, t)\|^2\big], \\ t \sim \mathcal{U}([0, 1]), \mathbf{x}_t \sim p_t. \tag{61}$$

However, this is (almost always) intractable, because we do not know the velocity field yet and we cannot sample $p_t$. In order to alleviate these issues, we can drastically simplify the loss by conditioning on the target sample $\mathbf{x}_1$:

$$\mathcal{L}_{\mathrm{CFM}}(\boldsymbol{\theta}) \triangleq \mathbb{E}_{t,\mathbf{x}_1,\mathbf{x}_t}\big[\|v_{\boldsymbol{\theta}}(\mathbf{x}_t, t) - v(\mathbf{x}_t, t \mid \mathbf{x}_1)\|^2\big], \\ t \sim \mathcal{U}([0, 1]), \mathbf{x}_1 \sim p_1, \mathbf{x}_t \sim p_{t|1}(\cdot \mid \mathbf{x}_1), \tag{62}$$

---

**Algorithm 3** Flow Matching training.

---

**Require:** Untrained velocity model $v_{\boldsymbol{\theta}}$, Schedule $\alpha_t, \beta_t : [0,1] \to [0,1]$, Source distribution $p$, Target distribution $q$.

1: **while** not converged **do**
2:     Sample $\mathbf{x}_0 \sim p$, $\mathbf{x}_1 \sim q$, and $t \sim \mathcal{U}([0,1])$.
3:     Compute $\mathbf{x}_t = \alpha_t \mathbf{x}_1 + \beta_t \mathbf{x}_0$ and $\mathbf{v}_t = \dot{\alpha}_t \mathbf{x}_1 + \dot{\beta}_t \mathbf{x}_0$.
4:     Compute the loss $\mathcal{L}_{\text{CFM}} = \|v_{\boldsymbol{\theta}}(\mathbf{x}_t, t) - \mathbf{v}_t\|^2$.
5:     Do an optimization step with $\nabla_{\boldsymbol{\theta}} \mathcal{L}_{\text{CFM}}$.
6: **end while**
7: **return** $v_{\boldsymbol{\theta}}$

---

where the conditional velocity is

$$v(\mathbf{x}, t \mid \mathbf{x}_1) = \dot{\alpha}_t \mathbf{x}_1 + \dot{\beta}_t \mathbf{x}_0, \quad \mathbf{x}_t = \mathbf{x} \tag{63}$$

$$= \dot{\alpha}_t \mathbf{x}_1 + \frac{\dot{\beta}_t}{\beta_t}(\mathbf{x} - \alpha_t \mathbf{x}_1) \tag{64}$$

$$= \left(\dot{\alpha}_t - \frac{\alpha_t \dot{\beta}_t}{\beta_t}\right)\mathbf{x}_1 + \frac{\dot{\beta}_t}{\beta_t}\mathbf{x}. \tag{65}$$

Remarkably, these two loss functions have the same gradient w.r.t. the parameters (Lipman et al., 2023):

$$\nabla_{\boldsymbol{\theta}} \mathcal{L}_{\text{FM}} = \nabla_{\boldsymbol{\theta}} \mathcal{L}_{\text{CFM}}. \tag{66}$$

This justifies applying gradient-based optimization methods on the conditional loss, which is tractable, because it will lead to the same parameter updates. See Algorithm 3 for the training algorithm. Refer to (Lipman et al., 2024) for an in-depth treatment of flow matching models.

Lastly, instead of sampling from a deterministic ODE, we can also consider sampling from a family of SDEs:

$$\mathrm{d}\mathbf{x}_t = \left(v(\mathbf{x}_t, t) + \frac{\sigma^2(t)}{2\eta_t}(v(\mathbf{x}_t, t) - \kappa_t \mathbf{x}_t)\right)\mathrm{d}t + \sigma(t)\,\mathrm{d}B_t, \tag{67}$$

where $B_t$ is a Brownian motion, $\sigma : [0,1] \to \mathbb{R}^{d \times d}$ is an arbitrary state-independent diffusion coefficient, and

$$\eta_t \triangleq \beta_t \left(\frac{\dot{\alpha}_t}{\alpha_t}\beta_t - \dot{\beta}_t\right), \quad \kappa_t \triangleq \frac{\dot{\alpha}_t}{\alpha_t}. \tag{68}$$

It can be shown that the generative processes in Equation (56) and Equation (67) have equivalent time marginals (Maoutsa et al., 2020). In the memoryless noise schedule, we have $\sigma(t) = \sqrt{2\eta_t}$ (Domingo-Enrich et al., 2025).

## F.2 DIFFUSION MODELS

Diffusion models take a (slightly) different perspective than flow matching. They view sampling as the reversal of a data destruction (or noising) process. For this, we must first define the noising process:

$$\mathbf{x}_{t+1} = \sqrt{\gamma_t}\mathbf{x}_t + \sqrt{1 - \gamma_t}\boldsymbol{\epsilon}_t, \quad \boldsymbol{\epsilon}_t \sim \mathcal{N}(\mathbf{0}, \mathbf{I}_n), \tag{69}$$

where $\gamma_t$ follows some schedule from 0 to $T$ such that $\mathbf{x}_T \sim \mathcal{N}(\mathbf{0}, \mathbf{I}_n)$.[1] As such, starting from $\mathbf{x}_0 = \mathbf{x} \sim q$, the data gets progressively more like Gaussian noise. Using Gaussian arithmetic, the above process can be computed in a closed form:

$$\mathbf{x}_t = \sqrt{\bar{\gamma}_t}\mathbf{x}_0 + \sqrt{1 - \bar{\gamma}_t}\boldsymbol{\epsilon}, \quad \boldsymbol{\epsilon} \sim \mathcal{N}(\mathbf{0}, \mathbf{I}_n), \tag{70}$$

where $\bar{\gamma}_t = \prod_{s=0}^{t-1} \gamma_s$. The denoising process from time $T$ to 0 can be computed as follows:

$$\mathbf{x}_{t-1} = \frac{1}{\sqrt{\gamma_t}}\left(\mathbf{x}_t - \frac{1 - \gamma_t}{\sqrt{1 - \bar{\gamma}_t}}\boldsymbol{\epsilon}_t\right) + \sigma_t \mathbf{z}, \quad \mathbf{z} \in \mathcal{N}(\mathbf{0}, \mathbf{I}_d). \tag{71}$$

Here the only unknown is $\boldsymbol{\epsilon}_t$, so we will train a network to approximate it; see Algorithm 4

---

[1]Generally, $(\gamma_t, \bar{\gamma}_t)$ are denoted by $(\alpha_t, \bar{\alpha}_t)$. This notation is used here to avoid confusion with flow matching schedules.

---

**Algorithm 4** Diffusion model training.

---

**Require:** Untrained epsilon model $\epsilon_{\boldsymbol{\theta}}$, Schedule $\{\gamma_t\}_{t=0}^T$, Target distribution $q$
  1: **while** not converged **do**
  2:     Sample $\mathbf{x}_0 \sim q$, $\boldsymbol{\epsilon} \sim \mathcal{N}(\mathbf{0}, \mathbf{I}_n)$ and $t \sim \mathcal{U}([T])$.
  3:     Compute $\mathbf{x}_t = \sqrt{\bar{\gamma}_t}\mathbf{x}_0 + \sqrt{1 - \bar{\gamma}_t}\boldsymbol{\epsilon}$.
  4:     Compute the loss $\mathcal{L}_{\mathrm{DM}} = \|\epsilon_{\boldsymbol{\theta}}(\mathbf{x}_t, t) - \boldsymbol{\epsilon}\|^2$.
  5:     Do an optimization step with $\nabla_{\boldsymbol{\theta}}\mathcal{L}_{\mathrm{DM}}$.
  6: **end while**
  7: **return** $\epsilon_{\boldsymbol{\theta}}$

---

### F.3 DIFFUSION MODELS AS AN INSTANCE OF FLOW MATCHING

We can sample a diffusion model with the DDIM schedule through the following SDE (Domingo-Enrich et al., 2025):

$$d\mathbf{x}_t = \left( \frac{\dot{\bar{\gamma}}_t}{2\bar{\gamma}_t}\mathbf{x}_t - \left( \frac{\dot{\bar{\gamma}}_t}{2\bar{\gamma}_t} + \frac{\sigma^2(t)}{2} \right) \frac{\epsilon(\mathbf{x}_t, t)}{\sqrt{1 - \bar{\gamma}_t}} \right) dt + \sigma(t)\, dB_t. \tag{72}$$

In order to consolidate diffusion models and flow matching models into a common framework where $p_0 = \mathcal{N}(\mathbf{0}, \mathbf{I}_d)$, we will be working with the score function:

$$s(\mathbf{x}, t) \triangleq \nabla_{\mathbf{x}} \log p_t(\mathbf{x}) \tag{73}$$

$$s(\mathbf{x}, t) = \frac{1}{\eta_t}(v(\mathbf{x}, t) - \kappa_t \mathbf{x}) \tag{74}$$

$$s(\mathbf{x}, t) = -\frac{\epsilon(\mathbf{x}, t)}{\sqrt{1 - \bar{\gamma}_t}}. \tag{75}$$

We can now sample either a diffusion model or flow matching model by converting their parametrization to the score function and sampling the following SDE:

$$d\mathbf{x}_t = \left( \kappa_t \mathbf{x}_t + \left( \frac{\sigma^2(t)}{2} + \eta_t \right) s(\mathbf{x}_t, t) \right) dt + \sigma(t)\, dB_t, \tag{76}$$

In the case of diffusion models, we have

$$\alpha_t = \sqrt{\bar{\gamma}_t}, \quad \beta_t = \sqrt{1 - \bar{\gamma}_t} \tag{77}$$

with associated time derivatives:

$$\dot{\alpha}_t = \frac{\dot{\bar{\gamma}}_t}{2\sqrt{\bar{\gamma}_t}}, \quad \dot{\beta}_t = -\frac{\dot{\bar{\gamma}}_t}{2\sqrt{1 - \bar{\gamma}_t}}. \tag{78}$$

We will use the convention of flow matching models. Generally, in diffusion models, we have that time is discrete from $0$–$K$ and decreases when sampling. Thus, we have the following conversion between the two conventions:

$$\bar{\gamma}_t = \bar{\gamma}\lfloor K(1 - t) \rfloor \tag{79}$$

$$\dot{\bar{\gamma}}_t = K \cdot (\bar{\gamma}\lfloor K(1 - t) - 1 \rfloor - \bar{\gamma}\lfloor K(1 - t) \rfloor). \tag{80}$$

One can easily verify that this is equivalent to sampling from DDIM.

