# OpenReview forum: "Value Matching: Scalable and Gradient-Free Reward-Guided Flow Adaptation"
_ICLR.cc/2026/Conference — ICLR 2026 Poster_

### Official Review · Reviewer_U1zm · 2025-10-22

**Soundness:** 3
**Presentation:** 2
**Contribution:** 3
**Rating:** 4
**Confidence:** 4

**Summary:**

This work proposes an online algorithm called Value Matching (VM) to learn a value function within an optimal control framework, addressing the scalability challenges faced when adapting large-scale flow and diffusion models to downstream tasks via reward optimization. VM enables flexible control over the complexity of the value network, offering adjustable memory and computational requirements. It supports optimization with non-differentiable rewards and operates in an on-policy manner, allowing it to go beyond the training data distribution and explore high-reward regions. Experiments on image generation and molecular design tasks demonstrate that VM achieves better stability and sample efficiency compared to Classifier Guidance (CG).

**Strengths:**

1.It learns the scalar value function directly rather than estimating guidance gradients, leading to more stable training.
2.It offers significant theoretical and practical value by enabling low-cost reinforcement learning (RL) fine-tuning.

**Weaknesses:**

1.The evaluation is insufficient, with comparisons limited to only CT-PPO.
2.The core contribution is primarily encapsulated in Equation 11, but the paper devotes excessive space to background, making it hard to follow.

**Questions:**

1.Please clarify how your work differs from the following approaches, and discuss the performance gap:
Inference-Time Alignment Control for Diffusion Models with Reinforcement Learning Guidance
Efficient Controllable Diffusion via Optimal Classifier Guidance
2.Although your method reduces fine-tuning costs, can it outperform the following full fine-tuning approaches?
Large-scale Reinforcement Learning for Diffusion Models
Training Diffusion Models with Reinforcement Learning
DPOK: Reinforcement Learning for Fine-tuning Text-to-Image Diffusion Models

---

> ### Author Response · Authors · 2025-11-22
>
> We thank the Reviewer for recognizing our work as offering significant theoretical and practical value. In the following, we address several points and questions mentioned within the review.
>
> **1 Clarity**
>
> > The core contribution is primarily encapsulated in Equation 11, but the paper devotes excessive space to background, making it hard to follow.
>
> We appreciate the Reviewer's focus on clarity. However, we do not consider Section 4 to be background, since it identifies value function learning as fitting the role for efficient and gradient-free reward adaptation. Nonetheless, we would welcome further suggestions to improve the writing. To improve readability, we have revised Section 3 (Lines 138–141) to more clearly state that we operate in a black-box setting where only function evaluations, not gradients, are available. This enables applications in chemistry and scientific discovery.
>
> **2 Limited Comparison With Inference-Time Schemes**
>
> > Please clarify how your work differs from the following approaches, and discuss the performance gap: Inference-Time Alignment Control for Diffusion Models with Reinforcement Learning Guidance Efficient Controllable Diffusion via Optimal Classifier Guidance
>
> Both methods are concurrent submissions to the same conference, and therefore we believe that they are not suitable for direct comparison. Instead, we conducted a direct comparison with SVDD [1], which is a recent gradient-free inference-time scheme. In the appendix of the updated paper, Table 7 shows that VM is vastly more efficient at inference, where SVDD is 40–600 times slower. Conceptually, this makes sense, since SVDD has a runtime complexity of $\mathcal{O}(TM(B+R))$, where $B$ is the cost of the base model and $R$ denotes the cost of evaluating the reward. In contrast, VM has the much more scalable runtime complexity $\mathcal{O}(T(B+G))$, where $G$ is the cost of evaluating $\nabla_x V_{\theta}(x, t)$. Furthermore, as shown in Figure 6 of the updated paper, VM achieves higher aesthetic rewards with comparable diversity at a fraction of the inference cost. This establishes that learning a value function is far more efficient than local reward optimization during sampling.
>
> **3 Limited Comparison With Fine-Tuning Methods**
>
> > Although your method reduces fine-tuning costs, can it outperform the following full fine-tuning approaches? Large-scale Reinforcement Learning for Diffusion Models Training Diffusion Models with Reinforcement Learning DPOK: Reinforcement Learning for Fine-tuning Text-to-Image Diffusion Models
>
> To address this, we significantly broadened our experimental evaluation to include direct comparisons with DDPO [2] and DPOK [3]. As shown in Figure 6 of the updated paper, VM consistently matches or exceeds the performance of both methods across a range of reward scales. A key result is that VM remains substantially more stable at high reward scales, where DDPO and DPOK frequently suffer mode collapse and severe drops in diversity. VM maintains the expected reward-diversity trade-off while being considerably more computationally efficient, further demonstrating VM as a robust and lightweight alternative to fine-tuning.
>
> **Conclusion**
>
> We thank the Reviewer again for their suggestions. We believe that the substantially expanded experiments address the raised concerns of the Reviewer. If these additions fully address your concerns, we kindly ask you to consider increasing your score.
>
> [1] Li, Xiner, et al. "Derivative-free guidance in continuous and discrete diffusion models with soft value-based decoding." arXiv preprint arXiv:2408.08252 (2024).
>
> [2] Black, Kevin, et al. "Training Diffusion Models with Reinforcement Learning." The Twelfth International Conference on Learning Representations. 2024.
>
> [3] Fan, Ying, et al. "Dpok: Reinforcement learning for fine-tuning text-to-image diffusion models." Advances in Neural Information Processing Systems 36 (2023): 79858-79885.

---

### Official Review · Reviewer_9Esi · 2025-10-25

**Soundness:** 1
**Presentation:** 3
**Contribution:** 1
**Rating:** 2
**Confidence:** 5

**Summary:**

This paper proposes value matching (VM), an online approach that relies on a small learned value network for guiding the model towards higher rewards. The proposed method is derived from optimal control theory and effectively learns a vector field network and a value network that simultaneously approximate the running cost. On downstream generation tasks, the proposed approach achieved better performance across multiple objective rewards.

**Strengths:**

- The paper's proposed method is built upon adjoint matching (AM), which is intuitive and easy to follow given the AM framework.
- The paper, built upon AM, also shares some desirable mathematical properties or guarantees.
- The proposed method has better running time compared to gradient-based AM models, but the performance comparison remains unclear (see weaknesses).

**Weaknesses:**

- Although the theoretical part in the paper seems solid to me, the experimental evaluations have **very limited baselines and focus only on small-scale datasets or reward functions** to convincingly demonstrate the effectiveness of the proposed approach.
  - **Regarding the number of baselines**. For each task, only the CFG and a single model were used as the baseline, even though many existing works were mentioned in the paper. For example, for image generation, there are gradient-based methods such as Adjoint Matching, SOC, FlowGrad [1], and DFlow [2], as well as other RL approaches such as ReFL [3] and [4] (gradient-free) and DRaFT [5] (gradient-based). For molecule generation, DFlow and OCFlow [6] achieved decent generations with almost perfect stability and validity. Current results do not fully demonstrate the superiority of the proposed approach over existing work.
  - **Regarding the evaluation metric and reward function**. The reward function used in the experiments in this paper is either a toy example from existing work or a non-standard metric not established in previous work on the same task, further weakening the paper's claims, as most metrics are not necessarily comparable.
    - For example, the standard task in previous flow matching guidance papers on image generation almost all focus on guiding text-to-image models like SD2/3 and use robust metrics like CLIP score, PickScore, or HPSv2 to prevent easily hacking the rewards (e.g., [6] demonstrated that the compression metric can be easily hacked to almost perfect but meaningless generations). For molecule generation, it also remains unclear why the authors did not follow the standard molecule generative modeling evaluation setup in [7] (uses CFG), DFlow, and OCFlow to evaluate the generation and compare the results with these existing works easily.
    - For molecule generation, it is widely known that quantum chemical calculations are either computationally expensive (ab initio methods) or highly inaccurate (empirical or semiempirical methods). It remains unclear what class the calculation method used in the experiments belongs to, how accurate and generalizable it is (GEOM-Drug is known to have some unreasonable configurations), and what the computational time is, which is crucial for reproducibility. In addition, crucial molecular properties like stability and validity were never mentioned or compared in the paper. Therefore, I am highly skeptical about why the authors did not follow the standard and easy approach in existing work but opted for a seemingly far more complex setup. I would highly discourage such an approach when comparing with baselines for fairness.

To summarize, for a paper emphasizing the scalability of the proposed method, I believe the existing experiments are, in contrast, limited in scope and poorly credible in supporting its fundamental claims.

- **The theoretical contributions in this paper are limited to me**. The core idea is almost identical to [VGG-Flow](https://openreview.net/forum?id=6MmOy2Ji8V). The scale of the experiments and the number of baselines in this paper fall significantly short of those in VGG-Flow, even if the latter is to be considered concurrent. In addition, there are existing works that have explored the role of the value function or its equivalent, the Q-function, for generative modeling, such as [8] and [9]. Despite different application domains, the underlying core ideas are pretty similar to me. Given that the theoretical results primarily come from the adjoint-matching paper, the theoretical contributions are limited.

- The method's scalability hinges on the value network being "significantly smaller than the base model." But what happens when the reward function is extremely complex? A small network may fail to accurately model the true value function, creating a new performance bottleneck. Is there a trade-off between VM's memory savings and its ability to represent a complex reward landscape? This weakness also echoes in the paper's limited, small-scale evaluation, as thoroughly mentioned in the first part.

- The algorithm learns by regressing the value network's predictions $V_{\theta}(x_t, t)$ onto a Monte Carlo estimate of the cost functional $\hat{J}$ (Eq. 12). This target $\hat{J}$ is based on a single sample trajectory and also depends on the current value function itself. This can be a very high-variance target, which is known to make value-based RL difficult to stabilize. The paper uses a weighting scheme, but the inherent stability of this learning process, especially for very long trajectories, is a potential concern.

- To sample from VM, one must run both the large base model and the (smaller) value network at every step to compute the guiding gradient $\nabla V_{\theta}$. While this is still much faster than other gradient-based inference-time optimization schemes, it's not "free" and adds computational overhead compared to using a single fine-tuned model. Other gradient-free approaches, such as ReFT and [6], should be benchmarked to support a more credible claim. Additionally, this issue may be coupled with the expressive power of the value net for more complex rewards mentioned above, and it may not be easy to find a balance.

[1] Liu, Xingchao, et al. "Flowgrad: Controlling the output of generative odes with gradients." Proceedings of the IEEE/CVF Conference on Computer Vision and Pattern Recognition. 2023.

[2] Ben-Hamu, Heli, et al. "D-flow: Differentiating through flows for controlled generation." arXiv preprint arXiv:2402.14017 (2024).

[3] Luong, Trung Quoc, et al. "Reft: Reasoning with reinforced fine-tuning." arXiv preprint arXiv:2401.08967 (2024).

[4] Fan, Jiajun, et al. "Online reward-weighted fine-tuning of flow matching with wasserstein regularization." The Thirteenth International Conference on Learning Representations. 2025.

[5] Clark, Kevin, et al. "Directly fine-tuning diffusion models on differentiable rewards." arXiv preprint arXiv:2309.17400 (2023).

[6] Wang, Luran, et al. "Training free guided flow matching with optimal control." arXiv preprint arXiv:2410.18070 (2024).

[7] Hoogeboom, Emiel, et al. "Equivariant diffusion for molecule generation in 3d." International conference on machine learning. PMLR, 2022.

[8] Zhang, Shiyuan, Weitong Zhang, and Quanquan Gu. "Energy-weighted flow matching for offline reinforcement learning." arXiv preprint arXiv:2503.04975 (2025).

[9] Alles, Marvin, et al. "FlowQ: Energy-Guided Flow Policies for Offline Reinforcement Learning." arXiv preprint arXiv:2505.14139 (2025).

**Questions:**

Please refer to the list of weaknesses above. In addition:
- VM is an *on-policy* algorithm (Algorithm 1), which means it discards past trajectories after each update. On-policy methods are generally known to be sample-inefficient. While the paper shows VM is more efficient than CG, how does its absolute sample efficiency compare to (hypothetical) offline value-based methods?

---

> ### Author Response · Authors · 2025-11-22
> **Official Comment by Authors (Part 1)**
>
> We thank the Reviewer for recognizing our work as intuitive and easy to follow. In the following, we address several points and questions mentioned within the review.
>
> **1 Problem Setting Clarification**
>
> > Regarding the number of baselines. For each task, only the CFG and a single model were used as the baseline, even though many existing works were mentioned in the paper. For example, for image generation, there are gradient-based methods such as Adjoint Matching, SOC, FlowGrad [1], and DFlow [2], as well as other RL approaches such as ReFL [3] and [4] (gradient-free) and DRaFT [5] (gradient-based). For molecule generation, DFlow and OCFlow [6] achieved decent generations with almost perfect stability and validity. Current results do not fully demonstrate the superiority of the proposed approach over existing work.
>
> The Reviewer states that VM builds upon Adjoint Matching (AM) [10]. We thank the Reviewer for raising this point. To clarify, VM and AM consider _fundamentally different problems_. AM assumes access to structural information, specifically gradients of the reward function, during training. This assumption on the reward rules out application areas that rely on non-differentiable rewards (e.g., chemistry and scientific discovery). In contrast, VM operates in a strictly black-box setting, where only scalar reward evaluations are available. Moreover, gradient access yields a substantial advantage. For these reasons, we view AM, SOCM, ReFL, and DRaFT as addressing a different problem setting. In order to clarify the problem setting that we consider, we have made this explicit in the revised manuscript (Lines 138–141).
>
> That said, we agree that the fine-tuning baselines in our work were limited. Therefore, we significantly broadened our experimental evaluation to include direct comparisons with DDPO [11] and DPOK [12]. As shown in Figure 6 of the updated paper, VM consistently matches or exceeds the performance of both methods across a range of reward scales. A key result is that VM remains substantially more stable at high reward scales, where DDPO and DPOK frequently suffer mode collapse and severe drops in diversity. VM maintains the expected reward-diversity trade-off while being considerably more computationally efficient, further demonstrating VM as a robust and lightweight alternative to fine-tuning.
>
> **2 Evaluation Metrics**
>
> > Regarding the evaluation metric and reward function. The reward function used in the experiments in this paper is either a toy example from existing work or a non-standard metric not established in previous work on the same task, further weakening the paper's claims, as most metrics are not necessarily comparable. For example, the standard task in previous flow matching guidance papers on image generation almost all focus on guiding text-to-image models like SD2/3 and use robust metrics like CLIP score, PickScore, or HPSv2 to prevent easily hacking the rewards (e.g., [6] demonstrated that the compression metric can be easily hacked to almost perfect but meaningless generations).
>
> We appreciate the Reviewer's concern and want to clarify that compression-based and aesthetic rewards are standard in prior work, such as [11, 12, 13]. Further, the metrics mentioned by the Reviewer are typically used in gradient-based text-to-image tasks, which is not the setting we consider in this work as we argue in the previous point.
>
> Regarding the claim that compression rewards are easily hacked. We do not evaluate a method based on reward values only, but rather we evaluate the trade-off between reward maximization and straying from the base model, which is in line with the actual objective of Equation (7). As shown in Figure 6, VM does a much better job of this than other methods.

---

> > ### Author Response · Authors · 2025-11-22
> > **Official Comment by Authors (Part 2)**
> >
> > **3 Molecular Experiments**
> >
> > > For molecule generation, it is widely known that quantum chemical calculations are either computationally expensive (ab initio methods) or highly inaccurate (empirical or semiempirical methods). It remains unclear what class the calculation method used in the experiments belongs to, how accurate and generalizable it is (GEOM-Drug is known to have some unreasonable configurations), and what the computational time is, which is crucial for reproducibility. In addition, crucial molecular properties like stability and validity were never mentioned or compared in the paper. Therefore, I am highly skeptical about why the authors did not follow the standard and easy approach in existing work but opted for a seemingly far more complex setup. I would highly discourage such an approach when comparing with baselines for fairness.
> >
> > We thank the Reviewer for pointing this out and appreciate the concerns. We would like to clarify several important points regarding the molecular setting:
> >  1. _Level of theory and accuracy_: The dataset GEOM-Drugs was generated using GFN2-xTB as its level of theory, and our dipole moment computations use the same simulator, ensuring methodological consistency. For completeness, we refer the Reviewer to Figure 13 of [14], which shows the correlation between GFN2-xTB predictions and ground truth dipole moments.
> >  2. _Stability and validity metrics_: We now include an experiment with QED as a reward, which reports stability and validity computed using RDKit [15]. Table 2 of the revised manuscript shows that VM improves QED while simultaneously improving atom stability, molecule stability, and validity.
> >  3. _Why not QM9_: While QM9 is the standard benchmark in most works, it is broadly recognized as a dataset with limited chemical diversity and overly idealized structures. In contrast, we make use of a base model trained on GEOM-Drugs, which is a much more realistic dataset of drug-like molecules.
> >  4. _On comparisons with baselines_: Lastly, due to computational constraints, all comparisons with other black-box reward adaptation methods are restricted to the CIFAR base model. Since most works make use of QM9, we cannot make direct comparisons to them in a molecular setting.
> >
> > Moreover, we report the training times and memory requirements in Figure 5 for reproducibility. In addition to this, we include Tables 5 and 6 in the revised manuscript that displays these values numerically.
> >
> > **4 Differences From VGG-Flow**
> >
> > > The core idea is almost identical to VGG-Flow. The scale of the experiments and the number of baselines in this paper fall significantly short of those in VGG-Flow, even if the latter is to be considered concurrent.
> >
> > The Reviewer states that VM is almost identical to VGG-Flow [16]. We believe this statement is factually wrong, and in the following we respectfully clarify the key conceptual and technical differences between VM and VGG-Flow:
> >  1. _Different paradigms_: VGG-Flow solves a high-dimensional PDE via minimizing its residual, alike a physics-informed neural network (PINN). VM is not a PINN method, since it does not make use of the HJB equation governing the value function. The two methods rely on entirely different training paradigms.
> >  2. _Requires reward gradients_: VGG-Flow explicitly requires reward gradients in its loss, whereas VM does not. As we argue in the first point, this places it in a fundamentally different problem setting that rules out application areas that rely on non-differentiable rewards (e.g., chemistry and scientific discovery).
> >  3. _No convergence guarantees_: VGG-Flow does not use a memoryless noise schedule and therefore cannot guarantee convergence to the optimal tilted distribution, as established in [10]. In contrast, VM does employ a memoryless noise schedule for provable convergence to the optimal tilted distribution.

---

> > > ### Author Response · Authors · 2025-11-22
> > > **Official Comment by Authors (Part 3)**
> > >
> > > **5 Relation to Offline RL Flow Methods**
> > >
> > > > In addition, there are existing works that have explored the role of the value function or its equivalent, the Q-function, for generative modeling, such as [8] and [9]. Despite different application domains, the underlying core ideas are pretty similar to me. Given that the theoretical results primarily come from the adjoint-matching paper, the theoretical contributions are limited.
> > >
> > > We appreciate the Reviewer's concern. To clarify, EFM [8] and FlowQ [9] operate in a related but distinct regime. Firstly, they use flow models to parameterize the actor policy in an RL problem, which differs from the reward adaptation setting that we consider in our work. Second, they are offline algorithms that fine-tune the actor flow model, whereas VM is online and does no fine-tuning, making it much more efficient and scalable. Furthermore, their algorithms are also quite different. EFM fine-tunes the flow model using a reward-weighted loss, similar in spirit to ORW-CFM-W2 [4]. FlowQ is an offline actor-critic approach using a flow model as the critic. In summary, the problem setting, objective, and algorithms are distinct.
> > >
> > > In addition, our paper provides two standalone theoretical contributions (Propositions 1 and 2) which establish (i) that the value function remains differentiable even when the reward is not, and (ii) that VM converges to the true value function in expectation. We want to clarify that these results do not originate from the Adjoint Matching paper [1].
> > >
> > > **6 Value Network Scaling Ablation**
> > >
> > > > The method's scalability hinges on the value network being "significantly smaller than the base model." But what happens when the reward function is extremely complex? A small network may fail to accurately model the true value function, creating a new performance bottleneck. Is there a trade-off between VM's memory savings and its ability to represent a complex reward landscape? This weakness also echoes in the paper's limited, small-scale evaluation, as thoroughly mentioned in the first part.
> > >
> > > We thank the Reviewer for this question and as suggested by the Reviewer, we have performed a thorough ablation over six value network sizes (0.5M–92M parameters), which can be found in Table 3 of the updated paper. The results show that smaller networks tend to perform better and are more stable, while larger ones exhibit higher variance, likely due to overfitting during the online updates. These results support our design choice to use lightweight value networks and suggest that the value learning problem is easier than learning the reward itself, since the value function represents a smoothed reward. Therefore, VM achieves effective reward adaptation with small auxiliary networks and a minimal memory footprint.
> > >
> > > **7 Variance of Monte Carlo Estimate**
> > >
> > > > The algorithm learns by regressing the value network's predictions $V_\theta(x_t, t)$ onto a Monte Carlo estimate of the cost functional $\hat{J}$ (Eq. 12). This target $\hat{J}$ is based on a single sample trajectory and also depends on the current value function itself. This can be a very high-variance target, which is known to make value-based RL difficult to stabilize. The paper uses a weighting scheme, but the inherent stability of this learning process, especially for very long trajectories, is a potential concern.
> > >
> > > We thank the Reviewer for highlighting this important point. While the single-sample Monte Carlo targets do have variance, we do not observe instability in practice. Across all direct comparisons, we find that VM matches or outperforms the gradient-free fine-tuning baselines considered. The weighting function helps normalize variance across timesteps, where early timesteps naturally have higher variance. Moreover, unlike in traditional RL, flow models have fixed length trajectories over time domain $[0, 1]$, avoiding the long-horizon variance issues of classical RL. Although not required for stable training, exploring variance-reduction techniques could have the potential to further improve the performance of VM.

---

> > > > ### Author Response · Authors · 2025-11-22
> > > > **Official Comment by Authors (Part 4)**
> > > >
> > > > **8 Inference-Time Overhead of VM**
> > > >
> > > > > To sample from VM, one must run both the large base model and the (smaller) value network at every step to compute the guiding gradient $\nabla V_{\vec{\theta}}$. While this is still much faster than other gradient-based inference-time optimization schemes, it's not "free" and adds computational overhead compared to using a single fine-tuned model.
> > > >
> > > > We thank the Reviewer for noting this. To address concerns about inference cost, we ran all models both with and without the value network and measured the runtime for generating a batch of 128 samples. The results, summarized in Table 4 of the updated paper, show that the overhead is small, only 1–30\% relative to the base model’s sampling time. This overhead is negligible compared to inference-time schemes.
> > > >
> > > > > Other gradient-free approaches, such as ReFT and [6], should be benchmarked to support a more credible claim. Additionally, this issue may be coupled with the expressive power of the value net for more complex rewards mentioned above, and it may not be easy to find a balance.
> > > >
> > > > ReFT is designed for autoregressive LLMs and is not applicable in our flow-based setting. OC-Flow [6] is an inference-time method that leverages reward gradients to guide generation at inference-time. Instead, we conducted a direct comparison with SVDD [13], which is a recent gradient-free inference-time scheme. In the appendix of the updated paper, Table 7 shows that VM is vastly more efficient at inference, where SVDD is 40–600 times slower. Conceptually, this makes sense, since SVDD has a runtime complexity of $\mathcal{O}(TM(B+R))$, where $B$ is the cost of the base model and $R$ denotes the cost of evaluating the reward. In contrast, VM has the much more scalable runtime complexity $\mathcal{O}(T(B+G))$, where $G$ is the cost of evaluating $\nabla_x V_{\theta}(x, t)$. Furthermore, as shown in Figure 6 of the updated paper, VM achieves higher aesthetic rewards with comparable diversity at a fraction of the inference cost. This establishes that learning a value function is far more efficient than local reward optimization during sampling.
> > > >
> > > > **9 Sample Efficiency**
> > > >
> > > > > VM is an on-policy algorithm (Algorithm 1), which means it discards past trajectories after each update. On-policy methods are generally known to be sample-inefficient. While the paper shows VM is more efficient than CG, how does its absolute sample efficiency compare to (hypothetical) offline value-based methods?
> > > >
> > > > We thank the Reviewer for raising this point. Indeed, VM is an on-policy method, which means trajectories are refreshed after every update. However, as discussed in the manuscript (Lines 205–215), on-policy sampling is not a weakness in our setting but a requirement for effective reward adaptation in flow models, as it enables learning beyond regions of high data availability. Offline trajectories drawn from the pre-trained distribution quickly become uninformative as the policy moves toward higher-reward regions, leading to severe sample inefficiency. A gap between offline and online algorithms has also been observed in prior work, e.g., the gap between offline and online algorithms in Figure 2(a) of [4].
> > > >
> > > > **Conclusion**
> > > >
> > > > We thank the Reviewer again for the careful reading of our work. We believe that the substantially expanded experiments address the raised concerns of the Reviewer. If these additions fully address your concerns, we kindly ask you to consider increasing your score.
> > > >
> > > > [10] Domingo-Enrich, Carles, et al. "Adjoint Matching: Fine-tuning Flow and Diffusion Generative Models with Memoryless Stochastic Optimal Control." The Thirteenth International Conference on Learning Representations. 2025.
> > > >
> > > > [11] Black, Kevin, et al. "Training Diffusion Models with Reinforcement Learning." The Twelfth International Conference on Learning Representations. 2024.
> > > >
> > > > [12] Fan, Ying, et al. "Dpok: Reinforcement learning for fine-tuning text-to-image diffusion models." Advances in Neural Information Processing Systems 36 (2023): 79858-79885.
> > > >
> > > > [13] Li, Xiner, et al. "Derivative-free guidance in continuous and discrete diffusion models with soft value-based decoding." arXiv preprint arXiv:2408.08252 (2024).
> > > >
> > > > [14] Bannwarth, Christoph, Sebastian Ehlert, and Stefan Grimme. "GFN2-xTB—An accurate and broadly parametrized self-consistent tight-binding quantum chemical method with multipole electrostatics and density-dependent dispersion contributions." Journal of chemical theory and computation 15.3 (2019): 1652-1671.
> > > >
> > > > [15] RDKit: Open-source cheminformatics. https://www.rdkit.org
> > > >
> > > > [16] Liu, Zhen, et al. "Value Gradient Guidance for Flow Matching Alignment." The Thirty-ninth Annual Conference on Neural Information Processing Systems. 2025.

---

> > > > > ### Comment · Reviewer_9Esi · 2025-11-28
> > > > >
> > > > > I thank the authors for their detailed response, which partially solved my previous questions, but not all of them. Specifically, as the **evaluation metrics remain non-standard** and only **a limited number of baselines were compared**, I am not fully convinced by the practical performance improvement of the model. I will further elaborate on my justifications as follows.
> > > > >
> > > > > First, after reviewing the VGG-Flow paper more carefully, I do agree on the setup difference between VM and VGG-Flow -- the dependency on the reward gradient. However, I also noted that the underlying mathematical deduction is still, in principle, built upon the same methodology of leveraging the adjoint state (either in optimal control or adjoint matching), with the only difference being the usage of a proxy value net instead of a reward function. Though incremental in such a theoretical sense, the argument in the paper would have been significantly stronger if the authors had provided more convincing experimental results on widely used datasets/baselines.
> > > > >
> > > > > While the additional experimental results in the revised manuscript did better demonstrate the proposed method's performance, I believe the results are still not convincing enough.
> > > > >
> > > > > ## Regarding the T2I baseline
> > > > > I appreciate that the authors now include additional baseline results in Figure 6 and provide interesting interpretations. Nonetheless, my concerns regarding the rewards were not addressed. To be clear, I did not dismiss the results with the easy metrics like compressibility. I do notice that these metrics also appeared in previous work (as I have mentioned in my review). However, I emphasize that most previous works only regarded these metrics as toy examples and carried out additional experiments on more complex and robust metrics like CLIP score, PickScore, or HPSv2. Therefore, since the authors did not address my concerns, the **scalability of VM to more complex reward functions remains unclear**.
> > > > >
> > > > > This problem, as already mentioned in my previous review, is also closely related to new expressiveness results on the size of the value model. Intuitively, it is extremely hard to distill a complex model-based reward like the CLIP score into a small model, especially since the value model is time-dependent. However, for easy metrics, it seems possible (as demonstrated in the new Table 3). Given the current experimental results, I believe scalability to more complex rewards is probably going to be more difficult.
> > > > >
> > > > > ## Regarding the molecule generation baseline
> > > > > I thank the authors for their clarifications, and I do agree that non-differentiable rewards like semi-empirical physiochemical properties cannot be adapted to gradient-based approaches like D-Flow. My point is, by following the previous practice like E3Diffusion, we can obtain a direct comparison with the existing approaches. Indeed, **no single baseline was compared for this task (both gradient-based and gradient-free)**, despite their success in previous works. I do acknowledge that the proposed VM's improvements in the rewards (In the new Table 2, for example), but it remains unclear whether VM can also outperform other existing approaches, especially since it does not always outperform the other baseline on the T2I task.
> > > > >
> > > > > It is acceptable (actually expected) if VM does not outperform the gradient-based guidance approaches like D-Flow if such gradient information is available (and it is indeed available in most previous works' setup). My point is, such a direct comparison can provide crucial practical insight for the viewers to choose between a costly but higher reward or a more efficient but lower reward.
> > > > >
> > > > > Therefore, the authors did not address my major concerns regarding the practical performance and scalability of VM. Although I do acknowledge that the new results help better understand the performance of VM, they did not provide strong enough evidence. Therefore, I maintain my original overall score. (I would give 3 if there is an option.)

---

> ### Author Response · Authors · 2025-12-03
> **Official Comment by Authors (Part 1)**
>
> We thank the Reviewer for their detailed response. In the following, we provide our response.
>
> **Limited baselines and non-standard evaluation metrics**
>
> We appreciate the reviewer’s continued engagement. We would like to clarify that the evaluation metrics we use follow standard practice in gradient-free reward adaptation. In particular, the reward functions are identical to those used in DDPO [11], DPOK [12], and related work. In addition, we report FID as a measure of distributional shift from the pre-trained model and Vendi diversity as an established metric for sample diversity.
>
> Regarding baselines, we want to again emphasize that our goal was to compare against representative _gradient-free_ adaptation methods. We include three comparisons with three fine-tuning methods and show that VM achieves similar reward performance while requiring significantly less memory. For example, VM adapts Stable Diffusion 2 using only 15 GB VRAM, whereas fine-tuning-based methods typically require >200 GB. We also compare against the inference-time gradient-free method SVDD and observe higher performance with lower overhead.
>
> **Setup difference with VGG-Flow**
>
> We agree that several recent methods share the SOC perspective, but we do not believe that this fully characterizes any given method. Each method differs substantially in how they acquire the adjoint state. For example, Adjoint Matching relies on the adjoint ODE, while VGG-Flow uses the HJB equation and they both require reward gradients and model fine-tuning. Although Adjoint Matching and VGG-Flow both leverage the SOC perspective and learn the adjoint states, we do not consider them to be incremental works since they both provide new perspectives on how to learn the adjoint states.
>
> In the same way, our work provides a new perspective in that VM learns a value function online and obtains adjoint states by backpropagation at inference time, without fine-tuning the base model or requiring reward gradients. As demonstrated in our experiments, this yields a practical and efficient alternative to fine-tuning methods. Also, it allows for optimization of non-differentiable rewards, which is not possible with previous methods that learn the adjoint states directly due to them requiring a differentiable reward.
>
> To be clear, this difference from VGG-Flow is in addition to the fact that it uses a fundamentally different training paradigm than VM, i.e., a PINN-based method.
>
> **Text-to-image baseline**
>
> Whereas we consider VM and gradient-based methods to consider fundamentally different problem settings, we agree with the Reviewer that it "can provide crucial practical insight for readers to choose between a costly but higher reward or a more efficient but lower reward." As such, we provide a similar experiment to [10], where we use Stable Diffusion 1.5 as the base model and ImageReward as the reward function. (This is not the exact same setup because the pre-trained model of [10] is not publicly available.) We provide the same metrics for Stable Diffusion 1.5 and its VM-adapted version over 1K samples with a fixed seed. For adaptation, we use CFG scale 4 during training and inference.
>
> | Method | ImageReward | ClipScore | PickScore | HPSv2 | DreamSim Diversity |
> |--|--|--|--|--|--|
> | Base Model | 0.110 | 33.32 | 19.72 | 26.29 | 35.89 |
> | Value Matching | 0.239 | 33.51 | 19.79 | 26.42 | 36.01 |
>
> As shown above, VM provides moderate improvements over the base model. The gains are smaller than those reported for gradient-based methods in prior work (see Table 5 of [10]). As also noted by the Reviewer, we believe this difference is expected, because gradient-based approaches have access to substantially richer information.

---

> ### Author Response · Authors · 2025-12-03
> **Official Comment by Authors (Part 2)**
>
> **Molecule generation baseline**
>
> We appreciate the Reviewer pointing us to molecular generation baselines. These approaches (FlowGrad, D-Flow, OC-Flow, and E3Diffusion) address a related but different setting. They condition on specific values of molecular properties such as dipole moment, with the target values drawn from the pre-training data distribution. In contrast, our setup optimizes for _maximum_ dipole moment, which pushes the model outside the support of the dataset distribution. This distinction is important because the conditional methods mentioned above inherently rely on property targets that already exist in the data and therefore do not evaluate out-of-distribution optimization. Additionally, FlowGrad, D-Flow, and OC-Flow require reward gradients and are demonstrated only on the small QM9 dataset, which is far simpler and contains far less realistic molecules than GEOM-Drugs.
>
> Nevertheless, to provide a gradient-based comparison on this modality, we compare VM to Adjoint Matching on this reward. Because AM requires differentiable reward signals, we first train a predictive model of dipole moments on GEOM-Drugs. We note that this can lead to out-of-distribution errors when Adjoint Matching ultimately goes beyond the data distribution. In contrast, VM can directly query the GFN2-xTB simulator during optimization, ensuring accurate rewards, even outside the dataset distribution. Also, AM cannot enforce validity constraints or geometry relaxation, since they introduce non-differentiable transformations. VM does not suffer from this restriction.
>
> Despite these challenges, we report a full comparison below. Dipole moments (computed via GFN2-xTB after geometry relaxation with GFN-FF) are computed over subsets _(all) / (non-fragmented) / (valid)_ across 10K samples. We also include stability and validity metrics for completeness.
>
> | Method | Dipole Moment (Debye) ↑ | Atoms stable (%) ↑ | Mols stable (%) ↑ | Non-frag. (%) ↑ | Valid (%) ↑ |
> |--|--|--|--|--|--|
> | Base Model | 6.4/6.4/6.4 | 98.0 | 49.5 | 68.5 | 48.3 |
> | Adjoint Matching | 9.1/8.9/6.7 | 89.0 | 5.4 | 67.1 | 4.0 |
> | Value Matching | 7.7/7.5/7.5 | 92.7 | 12.6 | 72.4 | 33.8
>
> As shown in the table, Adjoint Matching produces higher dipole moments when all generated molecules are included. However, when we focus only on valid molecules, VM achieves higher dipole moments as well as better stability and validity. This happens because Adjoint Matching cannot handle non-differentiable steps like validity checks or geometry relaxation, which makes it more likely to exploit the reward by generating invalid or fragmented molecules. VM can simply assign zero reward to such cases, which keeps the optimization focused on chemically meaningful structures. This illustrates the importance of being able to use non-differentiable reward functions, which VM naturally supports.

---

### Official Review · Reviewer_UG8D · 2025-10-30

**Soundness:** 3
**Presentation:** 3
**Contribution:** 3
**Rating:** 4
**Confidence:** 4

**Summary:**

The paper proposes Value Matching (VM), a novel, scalable, and gradient-free algorithm for reward-guided adaptation of large-scale flow and diffusion models. VM uses less memory than finetuning methods and offers improved stability compared to classifier guidance.

**Strengths:**

1. By decoupling the value learning from the base model, VM requires a separate, small value network. This reduces memory usage by over 95% compared to fine-tuning.
2. VM operates on-policy, which gives it enhanced reward expressivity and stability compared to CG.
3. The presentation of this work is clear and easy to follow, with abundant theoretical justifications.

**Weaknesses:**

1. The main quantitative results in Figures 8 and 9 focus only on Reward, KL, and FID. Why not directly compare the generated samples' diversities?
2. The paper contrasts VM's simplicity with the "extensive hyperparameter search" required by CT-PPO. However, this work does not include essential ablations studies on the architecture and size of the separate value network. It is important to know the sensitivity of VM's final performance to variations in the value network. Does VM also require a careful choice of a value network to support acceptable performance?
3. A concern lies in the selection of inference-time techniques. While CG is a classic baseline, its performance and controllability are not comparable to other non-fine-tuning guidance methods. Why not compare with those in experiments?
4. Figure 6 does not compare with any baselines. I think any training or non-training methods can perform well for the compressibility task.
5. In Figure 9, why is VM only compared with CT-PPO? For example, for aesthetic scores, many direct propagation algorithms can easily achieve >7 aesthetic reward in 10 epochs, but they are not mentioned. The current results are also confusing. It seems unclear what this means "VM demonstrates performance comparable to CT-PPO but with more predictable and stable behavior."

**Questions:**

see above

---

> ### Author Response · Authors · 2025-11-22
> **Official Comment by Authors (Part 1)**
>
> We thank the Reviewer for recognizing our work as clearly presented and theoretically justified. In the following, we address several points and questions mentioned within the review.
>
> **1 Missing Diversity Metric**
>
> > The main quantitative results in Figures 8 and 9 focus only on Reward, KL, and FID. Why not directly compare the generated samples' diversities?
>
> We thank the Reviewer for this suggestion. In the revised manuscript, we now include a diversity metric in Figure 6 using the Vendi diversity [1] computed on CLIP encodings. This metric highlights that fine-tuning methods suffer from mode collapse through a large drop in diversity as the reward scale increases, whereas VM does not.
>
> **2 Value Network Scaling Ablation**
>
> > The paper contrasts VM's simplicity with the "extensive hyperparameter search" required by CT-PPO. However, this work does not include essential ablations studies on the architecture and size of the separate value network. It is important to know the sensitivity of VM's final performance to variations in the value network. Does VM also require a careful choice of a value network to support acceptable performance?
>
> We thank the Reviewer for this question and as suggested by the Reviewer, we have performed a thorough ablation over six value network sizes (0.5M–92M parameters), which can be found in Table 3 of the updated paper. The results show that smaller networks tend to perform better and are more stable, while larger ones exhibit higher variance, likely due to overfitting during the online updates. These results support our design choice to use lightweight value networks and suggest that the value learning problem is easier than learning the reward itself, since the value function represents a smoothed reward. Therefore, VM achieves effective reward adaptation with small auxiliary networks and a minimal memory footprint.
>
> **3 Limited Comparison With Inference-Time Schemes**
>
> > A concern lies in the selection of inference-time techniques. While CG is a classic baseline, its performance and controllability are not comparable to other non-fine-tuning guidance methods. Why not compare with those in experiments?
>
> We would like to clarify the distinction between training-time and inference-time approaches. CG, like VM, has a training phase and is therefore not considered an inference-time scheme. However, we agree with the Reviewer that a comparison to a recent inference-time method would be valuable. As such, we conducted a direct comparison with SVDD [3], which is a recent gradient-free inference-time scheme. In the appendix of the updated paper, Table 7 shows that VM is vastly more efficient at inference, where SVDD is 40–600 times slower. Conceptually, this makes sense, since SVDD has a runtime complexity of $\mathcal{O}(TM(B+R))$, where $B$ is the cost of the base model and $R$ denotes the cost of evaluating the reward. In contrast, VM has the much more scalable runtime complexity $\mathcal{O}(T(B+G))$, where $G$ is the cost of evaluating $\nabla_x V_{\theta}(x, t)$. Furthermore, as shown in Figure 6 of the updated paper, VM achieves higher aesthetic rewards with comparable diversity at a fraction of the inference cost. This establishes that learning a value function is far more efficient than local reward optimization during sampling.
>
> **4 Compression Reward**
>
> > Figure 6 does not compare with any baselines. I think any training or non-training methods can perform well for the compressibility task.
>
> We appreciate the Reviewer raising this concern. However, due to limited computational resources, we were unable to run large-scale fine-tuning baselines on the SD2 model. Instead, Figure 5 and 7 of the revised paper demonstrate that VM scales to these large models with only 12GB memory, which fine-tuning methods cannot match.
>
> Regarding the claim that any method can perform well on the compression rewards. We do not evaluate a method based on reward values only, but rather we evaluate the trade-off between reward maximization and straying from the base model, which is in line with the actual objective of Equation (7). As shown in Figure 6, VM does a much better job of this than other methods.

---

> ### Author Response · Authors · 2025-11-22
> **Official Comment by Authors (Part 2)**
>
> **5 Limited Comparison With Fine-Tuning Methods**
>
> > In Figure 9, why is VM only compared with CT-PPO? For example, for aesthetic scores, many direct propagation algorithms can easily achieve $>7$ aesthetic reward in 10 epochs, but they are not mentioned.
>
> We thank the Reviewer for this suggestion. We would like to highlight that VM and direct propagation algorithms consider _fundamentally different problem settings_. VM assumes only access to black-box function evaluations of the reward function, whereas direct propagation algorithms assume access to structural information in the form of gradients. This assumption on the reward rules out application areas that rely on non-differentiable rewards (e.g., chemistry and scientific discovery). Because of this fundamental difference, we do not compare VM with such methods. In order to clarify the problem setting that we consider, we have made this explicit in the revised manuscript (Lines 138–141).
>
> That said, we agree that fine-tuning baselines in our work was limited. Consequently, we significantly broadened our experimental evaluation to include direct comparisons with DDPO [1] and DPOK [2]. As shown in Figure 6 of the updated paper, VM consistently matches or exceeds the performance of both methods across a range of reward scales. A key result is that VM remains substantially more stable at high reward scales, where DDPO and DPOK frequently suffer mode collapse and severe drops in diversity. VM maintains the expected reward-diversity trade-off while being considerably more computationally efficient, further demonstrating VM as a robust and lightweight alternative to fine-tuning.
>
> > It seems unclear what this means "VM demonstrates performance comparable to CT-PPO but with more predictable and stable behavior."
>
> We thank the Reviewer for raising this concern about clarity. By more predictable and stable behavior, we mean that VM adheres to the expected trade-off where an increased reward scaling results in higher rewards, along with straying further from the base model (measured by FID). In our experiments, CT-PPO showed less predictable trends, as its reward and FID curves did not reflect the expected trade-off. We have updated the manuscript to make this more clear (Lines 365–377).
>
> **Conclusion**
>
> We thank the Reviewer again for their constructive feedback. We believe that the clarifications and substantially expanded experiments address the raised concerns of the Reviewer. If these additions fully address your concerns, we kindly ask you to consider increasing your score.
>
> [1] Friedman, Dan, and Adji Bousso Dieng. "The vendi score: A diversity evaluation metric for machine learning." arXiv preprint arXiv:2210.02410 (2022).
>
> [2] Li, Xiner, et al. "Derivative-free guidance in continuous and discrete diffusion models with soft value-based decoding." arXiv preprint arXiv:2408.08252 (2024).
>
> [3] Black, Kevin, et al. "Training Diffusion Models with Reinforcement Learning." The Twelfth International Conference on Learning Representations. 2024.
>
> [4] Fan, Ying, et al. "Dpok: Reinforcement learning for fine-tuning text-to-image diffusion models." Advances in Neural Information Processing Systems 36 (2023): 79858-79885.

---

### Official Review · Reviewer_1EuJ · 2025-11-04

**Soundness:** 3
**Presentation:** 2
**Contribution:** 2
**Rating:** 6
**Confidence:** 3

**Summary:**

The authors propose an online algorithm for learning the value function of the flow matching models by formulating the reward-guided generation as an optimal control problem. By drawing the analogy that classifier guidance is seen as offline value function learning, the method enables reward-guided generation without reward fine-tuning the base flow matching model. Specifically, it trains a separate value model as the value function. This admits flexible reward model design (in terms of both architectures and model sizes) and significantly reduces memory usage compared to directly fine-tuning the base model. The method achieves comparable performance with PPO.

**Strengths:**

+ The connection between CG and VM is intuitive (VM viewed as an online generalization of CG)
+ The results of VM achieving comparable performance with PPO on image and molecule generation tasks look promising.

**Weaknesses:**

+ One of the major benefits of using reward guidance over directly doing reward fine-tuning (if the reward is non-differentiable, one can use an approach similar to [1]) is the reduced memory footprint. However, one can adopt LoRA to effectively reduce the memory requirement of the latter. I suggest the author compare different LoRA setups to show the trade-off of memory usage vs. fine-tuned performance of the baseline method to better illustrate how the proposed method does in preservation performance while using much less memory.
+ Since the value model is decoupled from the base model, it would be good to perform a bit of a scaling study on the scale of the value model.
+ Optimization with more reward functions will make the results more convincing.
+ It would be good to add [1] to related work as it also incorporates value function learning for reward fine-tuning flow-matching models.

[1] Reward Fine-Tuning Two-Step Diffusion Models via Learning Differentiable Latent-Space Surrogate Reward, CVPR 2025

**Questions:**

See weaknesses

---

> ### Author Response · Authors · 2025-11-22
> **Official Comment by Authors (Part 1)**
>
> We thank the Reviewer for recognizing our work as promising and offering a substantially reduced memory footprint. In the following, we address several points and questions mentioned within the review.
>
> **1 Related Work**
>
> > It would be good to add [1] to related work as it also incorporates value function learning for reward fine-tuning flow-matching models.
>
> We thank the Reviewer for identifying this related work. We have included it in the Related Works section of the updated paper (Lines 500–502).
>
> **2 LoRA Fine-Tuning**
>
> > One of the major benefits of using reward guidance over directly doing reward fine-tuning (if the reward is non-differentiable, one can use an approach similar to [1]) is the reduced memory footprint. However, one can adopt LoRA to effectively reduce the memory requirement of the latter. I suggest the author compare different LoRA setups to show the trade-off of memory usage vs. fine-tuned performance of the baseline method to better illustrate how the proposed method does in preservation performance while using much less memory.
>
> We appreciate the Reviewer’s observation that a key advantage of our method is its substantially reduced memory footprint. To address the suggestion, we measured the memory requirements of applying LoRA to DDPO fine-tuning on Stable Diffusion 2:
>
> | Rank | Trainable params (M) | Memory (GB) |
> |---------:|--------------------------:|-----------------:|
> | Full     | 865.9                    | 199.4            |
> | 128      | 62.0                     | 143.1            |
> | 64       | 31.0                     | 142.6            |
> | 32       | 15.5                     | 142.4            |
> | 16       | 7.8                      | 142.3            |
> | 8        | 3.9                      | 142.3            |
> | 4        | 1.9                      | 142.2            |
>
>
> While LoRA indeed lowers memory consumption relative to full-parameter tuning, it does not resolve the fundamental scaling issue, where memory usage grows proportionally with the base model size.
>
> **3 Value Network Scaling Ablation**
>
> > Since the value model is decoupled from the base model, it would be good to perform a bit of a scaling study on the scale of the value model.
>
> We thank the Reviewer for this question and as suggested by the Reviewer, we have performed a thorough ablation over six value network sizes (0.5M–92M parameters), which can be found in Table 3 of the updated paper. The results show that smaller networks tend to perform better and are more stable, while larger ones exhibit higher variance, likely due to overfitting during the online updates. These results support our design choice to use lightweight value networks and suggest that the value learning problem is easier than learning the reward itself, since the value function represents a smoothed reward. Therefore, VM achieves effective reward adaptation with small auxiliary networks and a minimal memory footprint.

---

> ### Author Response · Authors · 2025-11-22
> **Official Comment by Authors (Part 2)**
>
> **4 Limited Evaluation**
>
> > Optimization with more reward functions will make the results more convincing.
>
> To address the Reviewer's concern, we have included an additional experiment with a QED reward function for molecular generation in the revised manuscript. Additionally, we have performed significantly broadened our experimental evaluation to include direct comparisons with DDPO [2] and DPOK [3]. As shown in Figure 6 of the updated paper, VM consistently matches or exceeds the performance of both methods across a range of reward scales. A key result is that VM remains substantially more stable at high reward scales, where DDPO and DPOK frequently suffer mode collapse and severe drops in diversity. VM maintains the expected reward-diversity trade-off while being considerably more computationally efficient, further demonstrating VM as a robust and lightweight alternative to fine-tuning.
>
> Moreover, we conducted a direct comparison with SVDD [4], which is a recent gradient-free inference-time scheme. In the appendix of the updated paper, Table 7 shows that VM is vastly more efficient at inference, where SVDD is 40–600 times slower. Conceptually, this makes sense, since SVDD has a runtime complexity of $\mathcal{O}(TM(B+R))$, where $B$ is the cost of the base model and $R$ denotes the cost of evaluating the reward. In contrast, VM has the much more scalable runtime complexity $\mathcal{O}(T(B+G))$, where $G$ is the cost of evaluating $\nabla_x V_{\theta}(x, t)$. Furthermore, as shown in Figure 6 of the updated paper, VM achieves higher aesthetic rewards with comparable diversity at a fraction of the inference cost. This establishes that learning a value function is far more efficient than local reward optimization during sampling.
>
> **Conclusion**
>
> We thank the Reviewer again for their feedback. We hope that our clarifications and extensive additional evaluation address the raised concerns. If these additions fully address your concerns, we kindly ask you to consider increasing your score.
>
>
> [2] Black, Kevin, et al. "Training Diffusion Models with Reinforcement Learning." The Twelfth International Conference on Learning Representations. 2024.
>
> [3] Fan, Ying, et al. "Dpok: Reinforcement learning for fine-tuning text-to-image diffusion models." Advances in Neural Information Processing Systems 36 (2023): 79858-79885.
>
> [4] Li, Xiner, et al. "Derivative-free guidance in continuous and discrete diffusion models with soft value-based decoding." arXiv preprint arXiv:2408.08252 (2024).

---

### Author Response · Authors · 2025-11-22

We thank all reviewers for their constructive feedback. Reviewers highlighted our method as practically valuable (U1zm), clearly presented (UG8D), promising (1EuJ), and theoretically justified (UG8D, U1zm). Among other things, we have significantly strengthened the paper with expanded experimental comparisons, a comprehensive value network scaling study, and improved molecular evaluations. We also include an additional full page of experiments and improved clarity throughout the manuscript. All revisions are highlighted in blue for easy identification. Below, we address common concerns shared by multiple reviewers.

**1 Limited Comparison With Fine-Tuning Methods**

We significantly broadened our experimental evaluation to include direct comparisons with DDPO [1] and DPOK [2]. As shown in Figure 6 of the updated paper, VM consistently matches or exceeds the performance of both methods across a range of reward scales. A key result is that VM remains substantially more stable at high reward scales, where DDPO and DPOK frequently suffer mode collapse and severe drops in diversity. VM maintains the expected reward-diversity trade-off while being considerably more computationally efficient, further demonstrating VM as a robust and lightweight alternative to fine-tuning.

**2 Inference-Time Overhead of VM**

To address concerns about inference cost, we ran all models both with and without the value network and measured the runtime for generating a batch of 128 samples. The results, summarized in Table 4 of the updated paper, show that the overhead is small, only 1–30\% relative to the base model’s sampling time. As shown next, this overhead is negligible compared to inference-time schemes.

**3 Limited Comparison With Inference-Time Schemes**

We conducted a direct comparison with SVDD [3], which is a recent gradient-free inference-time scheme. In the appendix of the updated paper, Table 7 shows that VM is vastly more efficient at inference, where SVDD is 40–600 times slower. Conceptually, this makes sense, since SVDD has a runtime complexity of $\mathcal{O}(TM(B+R))$, where $B$ is the cost of the base model and $R$ denotes the cost of evaluating the reward. In contrast, VM has the much more scalable runtime complexity $\mathcal{O}(T(B+G))$, where $G$ is the cost of evaluating $\nabla_x V_{\theta}(x, t)$. Furthermore, as shown in Figure 6 of the updated paper, VM achieves higher aesthetic rewards with comparable diversity at a fraction of the inference cost. This establishes that learning a value function is far more efficient than local reward optimization during sampling.

**4 Value Network Scaling Ablation**

We performed a thorough ablation over six value network sizes (0.5M–92M parameters), which can be found in Table 3 of the updated paper. The results show that smaller networks tend to perform better and are more stable, while larger ones exhibit higher variance, likely due to overfitting during the online updates. These results support our design choice to use lightweight value networks and suggest that the value learning problem is easier than learning the reward itself, since the value function represents a smoothed reward. Therefore, VM achieves effective reward adaptation with small auxiliary networks and a minimal memory footprint.

**Conclusion**

We believe these additions, clarifications, and new experiments directly address all major reviewer concerns and further highlight VM’s strengths: stability, efficiency, and scalability, compared to fine-tuning and inference-time methods.

[1] Black, Kevin, et al. "Training Diffusion Models with Reinforcement Learning." The Twelfth International Conference on Learning Representations. 2024.

[2] Fan, Ying, et al. ”Dpok: Reinforcement learning for fine-tuning text-to-image diffusion models.” Advances in Neural
Information Processing Systems. 2023.

[3] Li, Xiner, et al. "Derivative-free guidance in continuous and discrete diffusion models with soft value-based decoding." arXiv preprint arXiv:2408.08252 (2024).

---

### Meta-Review · Area_Chair_TAVQ · 2025-12-29

**Summary:**

Reviewers mainly concerned about two points. One was about the connections and the differences between this manuscript and other reward adaptation method under the stochastic optimal control perspective like VGG-flow [1]. The other one was about the evaluation and comparison with different setup and scenarios.

[1] Liu, Zhen, et al. "Value Gradient Guidance for Flow Matching Alignment." The Thirty-ninth Annual Conference on Neural Information Processing Systems. 2025.

**Reviewer Concerns:**

For the connections and differences between the proposed method and other method under the perspective under stochastic optimal control, after I work through the manuscript and author-reviewer discussion, I feel the proposed method is a supplementary to the established methods like Adjoint-Matching [1], VGG-flow [2], SVDD [3], etc. Specifically, adjoint-matching and VGG-flow requires the gradient of the reward function, and SVDD is basically a test-time adaptation method that can be computationally-expensive. The method proposed in this manuscript improves upon SVDD, that parameterizes the value function and demonstrates that we can use the gradient of this parametric value function to simulate the optimal control path based on Pontryagin’s minimum principle and match the value function with the simulation to optimize the value function and get reasonable results, which is indeed a novel information to the community.

[1] Domingo-Enrich, Carles, Michal Drozdzal, Brian Karrer, and Ricky TQ Chen. "Adjoint Matching: Fine-tuning Flow and Diffusion Generative Models with Memoryless Stochastic Optimal Control." In The Thirteenth International Conference on Learning Representations.

[2] Liu, Zhen, Tim Z. Xiao, Carles Domingo-Enrich, Weiyang Liu, and Dinghuai Zhang. "Value Gradient Guidance for Flow Matching Alignment." In The Thirty-ninth Annual Conference on Neural Information Processing Systems.

[3] Li, Xiner, Yulai Zhao, Chenyu Wang, Gabriele Scalia, Gokcen Eraslan, Surag Nair, Tommaso Biancalani et al. "Derivative-free guidance in continuous and discrete diffusion models with soft value-based decoding." arXiv preprint arXiv:2408.08252 (2024).

Regarding the empirical evaluation, the authors conduct additional experimental results and resolve most the questions on the comparison with different baselines and empirical criterion. There are still concerns on if the proposed methods can still have good performances when the reward function is complex. Given there are limited baseline on gradient-free guidance with complex reward function, I think this can be a more general question that can leave for further study.

**Reviewer Scores:**

Reviewer 9Esi may be increased their score as their main focus was about the empirical evaluation and the relationship between the proposed method and VGG-flow. The reviewer agreed that the proposed method focused on different problems. Although the reviewer still has some concern on the empirical evaluation, I think the authors address the majority of the concern.

Reviewer UG8D mainly focused on the evaluation criterion and the missing baselines, and I believe the authors provide sufficient supplementary experimental results that justify the effectiveness of the proposed method.

Reviewer U1zm mainly had some questions about the comparison with some baseline methods, which I believe the authors already solve it.

---

### Decision · Program_Chairs · 2026-01-26

Accept (Poster)